# Non-canonical cMet regulation by vimentin mediates Plk1 inhibitor–induced apoptosis

Ratnakar Singh[1] iD, Shaohua Peng[1], Pavitra Viswanath[1,2], Vaishnavi Sambandam[1], Li Shen[3], Xiayu Rao[3], Bingliang Fang[4], Jing Wang[2,3] & Faye M Johnson[1,2,*] iD

## Abstract

To address the need for improved systemic therapy for non–small-cell lung cancer (NSCLC), we previously demonstrated that mesenchymal NSCLC was sensitive to polo-like kinase (Plk1) inhibitors, but the mechanisms of resistance in epithelial NSCLC remain unknown. Here, we show that cMet was differentially regulated in isogenic pairs of epithelial and mesenchymal cell lines. Plk1 inhibition inhibits cMet phosphorylation only in mesenchymal cells. Constitutively active cMet abrogates Plk1 inhibitor–induced apoptosis. Likewise, cMet silencing or inhibition enhances Plk1 inhibitor–induced apoptosis. Cells with acquired resistance to Plk1 inhibitors are more epithelial than their parental cells and maintain cMet activation after Plk1 inhibition. In four animal NSCLC models, mesenchymal tumors were more sensitive to Plk1 inhibition alone than were epithelial tumors. The combination of cMet and Plk1 inhibition led to regression of tumors that did not regrow when drug treatment was stopped. Plk1 inhibition did not affect HGF levels but did decrease vimentin phosphorylation, which regulates cMet phosphorylation via β1-integrin. This research defines a heretofore unknown mechanism of ligand-independent activation of cMet downstream of Plk1 and an effective combination therapy.

**Keywords** cMet; drug combination; NSCLC; Plk1; vimentin
**Subject Categories** Cancer; Pharmacology & Drug Discovery; Respiratory System

## Introduction

Lung cancer remains the most lethal cancer, contributing to 27% of cancer-related deaths in the United States (Siegel *et al*, 2018). Although personalized treatment approaches based on genetic mutations that underlie non–small-cell lung cancer (NSCLC) have

been developed and immunotherapy is very effective in some patients, the 5-year overall survival rate for all stages of NSCLC is only 18% and a dismal 5% for those with metastatic disease (Siegel *et al*, 2018). Therefore, numerous new potential targets, including several cell cycle kinases, are under investigation to further improve patient survival. Polo-like kinase 1 (Plk1) is a serine–threonine protein kinase that is overexpressed in cancer cells and plays a major role in the regulation of G2–M transition and response to DNA damage. Inhibitors of Plk1 are under active clinical development in oncology (NCT03414034 and NCT03303339).

Despite Plk1's reputation as an essential protein for cell survival, Plk1 inhibitors are well tolerated by patients (Nokihara *et al*, 2016; Pujade-Lauraine *et al*, 2016; Schoffski *et al*, 2012, 2010; Sebastian *et al*, 2010; Stadler *et al*, 2014; Van den Bossche *et al*, 2016) and there are diverse biological responses to Plk1 inhibition or knockdown in cancer cells (Choi *et al*, 2015; Craig *et al*, 2014; Driscoll *et al*, 2014; Gjertsen & Schoffski, 2015; McCarroll *et al*, 2015; Medema *et al*, 2011; Rudolph *et al*, 2009; Spankuch-Schmitt *et al*, 2002). These laboratory results are consistent with the results of clinical trials of Plk1 inhibitors in solid tumors that demonstrated striking clinical responses but a low response rate (4–14%), with a stable disease rate of 26–42% in unselected patients (Nokihara *et al*, 2016; Pujade-Lauraine *et al*, 2016; Schoffski *et al*, 2012, 2010; Sebastian *et al*, 2010; Stadler *et al*, 2014; Van den Bossche *et al*, 2016). Up to 11% of patients had stable disease for more than a year (Pujade-Lauraine *et al*, 2016; Sebastian *et al*, 2010). Predictive biomarkers have not been used to select patients likely to respond to Plk1 inhibitors. Plk1 inhibitor sensitivity was found to be associated with *KRAS* and *TP53* mutations in colon, breast, and lung tumors in some studies (Degenhardt *et al*, 2010; Luo *et al*, 2009; Sanhaji *et al*, 2013) but not in others (Ferrarotto *et al*, 2016; Sanhaji *et al*, 2012). We previously compared gene mutation and basal gene and protein expression among 63 NSCLC cell lines and discovered that mesenchymal NSCLC cell lines were more sensitive to Plk1 inhibitors than were epithelial cell lines *in vitro* and *in vivo*; however, *KRAS*, *TP53*, and *MET* mutations did not consistently

1 Department of Thoracic/Head & Neck Medical Oncology, The University of Texas MD Anderson Cancer Center, Houston, TX, USA
2 The University of Texas MD Anderson Cancer Center Graduate School of Biomedical Sciences, Houston, TX, USA
3 Department of Bioinformatics and Computational Biology, The University of Texas MD Anderson Cancer Center, Houston, TX, USA
4 Department of Thoracic and Cardiovascular Surgery, The University of Texas MD Anderson Cancer Center, Houston, TX, USA
*Corresponding author. Tel: +1 713-792-6363; Fax: +1 713-792-1220; E-mail: fmjohns@mdanderson.org
Dr. Ratnakar Singh is currently a postdoctoral fellow at the University of Illinois. Ms. Pavitra Visvanath is currently a graduate student at the University of Utah. Both were employed at The University of Texas MD Anderson Cancer Center at the time this research was performed

predict sensitivity. However, only one NSCLC cell line in the analysis had an activating mutation in exon 14 of *MET* making it impossible to determine whether this molecular subgroup was resistant to Plk1 inhibition. Plk1 inhibitors were equally effective at inhibiting Plk1 in mesenchymal/sensitive and epithelial/resistant NSCLC cell lines (Ferrarotto *et al*, 2016). The mechanisms of resistance to Plk1 inhibitors in epithelial NSCLC remain unknown, and this represents a major gap in knowledge.

To address this gap, in the current study, we performed an integrated analysis of functional proteomics and drug screening in an independent online database of NSCLC cell lines. When this approach confirmed that epithelial-to-mesenchymal transition (EMT)-related proteins correlated significantly with Plk1 inhibitor sensitivity, we used isogenic pairs of epithelial NSCLC cell lines treated with TGF-β to induce a mesenchymal phenotype to measure the changes in protein expression and activation after Plk1 inhibition. We observed differential regulation of cMet phosphorylation after Plk1 inhibition in epithelial and mesenchymal NSCLC. We confirmed cMet's role in Plk1 inhibition–induced apoptosis by inhibiting, silencing, and activating cMet in NSCLC *in vivo* and *in vitro*. Further, Plk1 inhibition decreases vimentin phosphorylation that subsequently regulates cMet phosphorylation via β1-integrin only in mesenchymal NSCLC.

# Results

## NSCLC cell lines with high cMet and epithelial protein expression are resistant to Plk1 inhibitors *in vitro*

To find pathways that can drive Plk1 inhibitor resistance, we examined data for all NSCLC cell lines from two sources: (i) protein and phosphoprotein expression measured using reverse phase protein array (RPPA) from the MD Anderson Cell Line Project database (Li *et al*, 2017), and (ii) Plk1 inhibitor sensitivity from the Cancer Therapeutics Response Portal v2 (CTRPv2; https://portals.broadinstitute.org/ctrp/) database (Seashore-Ludlow *et al*, 2015). These data were independent of our original study (Ferrarotto *et al*, 2016), although there was some overlap in the drugs and cell lines tested (Appendix Table S1 and Fig S1). Varying numbers of proteins were associated with sensitivity to four Plk1 inhibitors (selected by Spearman's rho coefficient value > 0.3; associated *P*-values are indicated in the figures): 33 proteins were associated with sensitivity to BI2536, 36 with sensitivity to GSK461364, 37 with sensitivity to BRD-K70511574, and 26 with sensitivity to GW-843682X (Fig EV1A–D). Thirty-three proteins were associated with sensitivity to two or more Plk1 inhibitors (Fig 1A). Consistent with our previous findings, we observed that expression of the epithelial proteins E-cadherin ($P < 0.001$) and β-catenin ($P < 0.001$) was higher and expression of the mesenchymal protein Snail ($P < 0.01$), as well as ATM and thymidylate synthase, was lower in cell lines resistant to Plk1 inhibitors than in those sensitive to Plk1 inhibitors. We also found that cMet protein expression correlated with drug sensitivity for all Plk1 inhibitors ($P < 0.01$; Figs 1B and EV1E).

The correlation of drug sensitivity with cMet protein expression motivated us to compare Plk1 inhibitor sensitivity [area under the curve (AUC) and effective dose 50 ($ED_{50}$)] to *MET* gene copy number in NSCLC cell lines. *MET* gene copy number was obtained from the MD Anderson Cell Line Project database, CTRPv2, and

Kubo *et al* (2009) in 41, 185, and 29 NSCLC cell lines, respectively. *MET* gene copy number did not correlate with drug sensitivity for any of the 24 possible comparisons (i.e., two measures of drug sensitivity, four drugs, and three sources of *MET* copy number) with Spearman's rho coefficient values that ranged from −0.428 to 0.430 and associated *P*-values that ranged from 0.078 to 0.872. However, this analysis was limited by the fact that there were drug sensitivity data for only two NSCLC cell lines with *MET* copy number > 5.

## Induction of a mesenchymal phenotype increases Plk1 inhibition–induced apoptosis

To create isogenic cell line pairs for mechanistic studies, we incubated epithelial/resistant NSCLC cells (H1975, HCC366, and HCC4006) with 5 ng/ml TGF-β for at least 14 days, which led to the expected changes in the expression of vimentin, Snail, Slug, ZEB1, Twist, E-cadherin, β-catenin, and claudin 7 (Fig 2A and Appendix Fig S2). Given that gene mutation did not correlate with Plk1 inhibitor sensitivity (Ferrarotto *et al*, 2016), we chose these cell lines independent of gene mutation status. The induction of a mesenchymal phenotype by TGF-β led to a significant increase in cleaved poly(ADP-ribose) polymerase (PARP) protein expression after Plk1 inhibition with volasertib in all isogenic cell lines (Fig 2B and Appendix Fig S3A). Similarly, TGF-β–treated mesenchymal cells showed increases in volasertib-induced apoptosis as measured by BrdU-positive cells (threefold in H1975, 4.1-fold in HCC4006, and 4.1-fold in HCC366) compared with their epithelial parental cells ($P < 0.05$; Fig 2C). Likewise, volasertib-induced DNA damage was increased in the mesenchymal cell lines compared with the epithelial parental cells (Fig 2B and D, and Appendix Fig S3A), and mesenchymal cell lines were more sensitive to volasertib *in vitro* (Appendix Fig S3B). The Plk1 inhibitor–induced DNA damage (Driscoll *et al*, 2014; Wang *et al*, 2018; Yim & Erikson, 2009) may explain why apoptosis measured by BrdU was more striking than that measured by PARP cleavage. Target inhibition, measured by inhibition of the Plk1 substrate p-NPM (S4), was similar in the isogenic pairs (Appendix Fig S3A).

## Activation of cMet is differentially regulated in epithelial and mesenchymal NSCLC cell lines following Plk1 inhibition and knockdown

To elucidate the mechanism for EMT-induced sensitivity to Plk1 inhibition, we analyzed the expression of 301 proteins or phosphoproteins (Kalu *et al*, 2017) after Plk1 inhibition with volasertib for 24 h using RPPA in the three isogenic pairs and two additional mesenchymal/sensitive cell lines (Calu6 and H1792; Fig 3A). To discover the pathway responsible for driving the resistance to Plk1 inhibition, we looked for differentially expressed proteins between epithelial and mesenchymal cell lines after treatment with volasertib. At a cutoff of $P < 0.05$, we observed that the mean expression of 33 proteins or phosphoproteins was differentially regulated (Fig 3B). These data revealed differential effects on both the cMet/FAK/Src axis and the PI3K/Akt pathway (Figs 3B and EV2A). We did a similar analysis using only the parental cell lines to eliminate TGF-β as a confounder, and this led to similar findings (Fig EV2B and C). Western blot analysis confirmed a significant decrease in cMet (Y1234/1235) phosphorylation in sensitive/mesenchymal cells

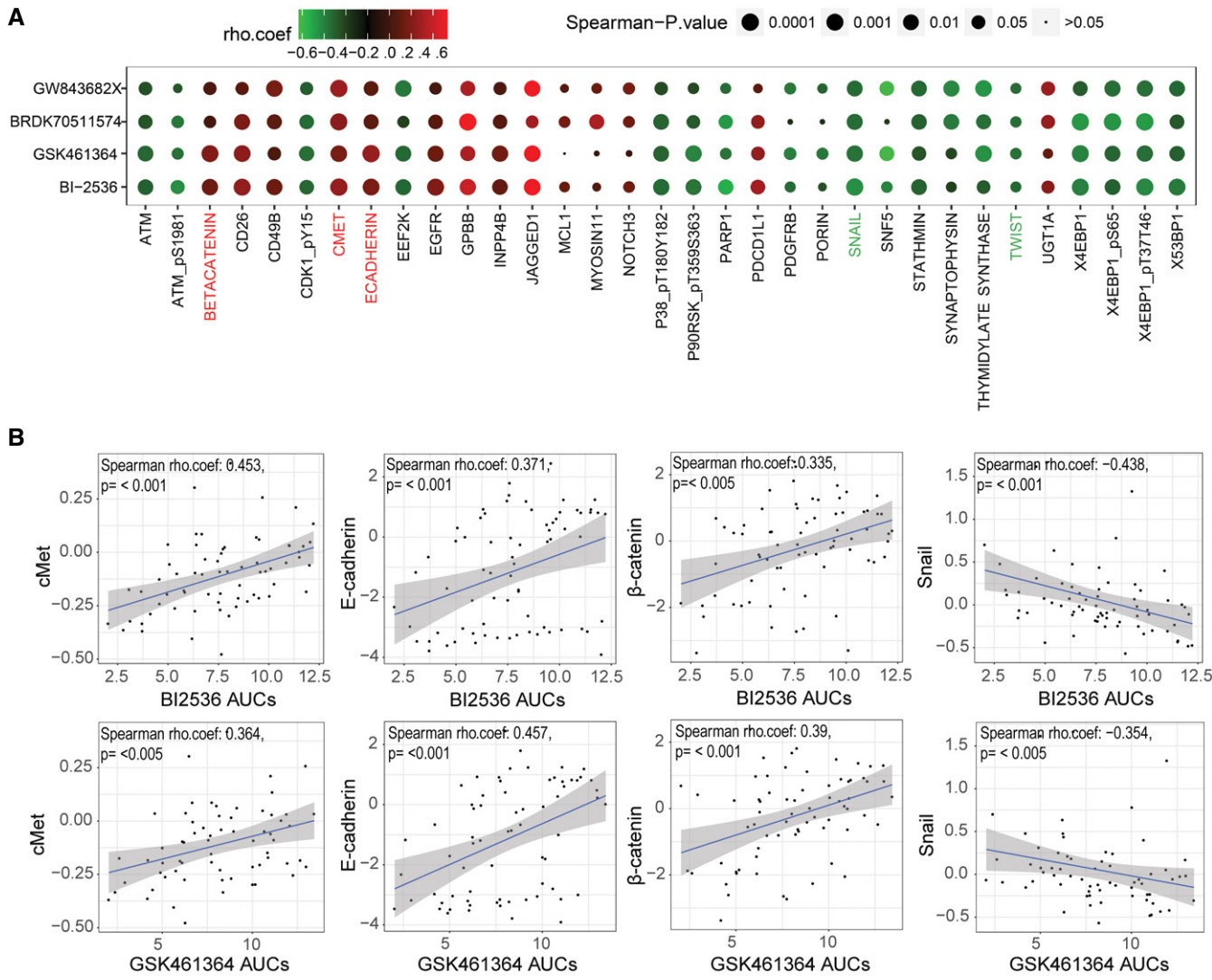

**Figure 1. Epithelial-to-mesenchymal transition proteins are associated with sensitivity to Plk1 inhibitors in non–small-cell lung cancer cell lines *in vitro* in an independent dataset.**

Spearman's correlations between protein expression and sensitivity to Plk1 inhibitors (BI2536, GSK461364, BRD-K70511574, and GW-843682X), based on data from the Cancer Therapeutics Response Portal v2 database and protein expression data derived from the MD Anderson Cell Line Project database (Li *et al*, 2017).

A   Proteins with expression levels that correlate with area under the curve (AUC) values for at least two drugs are included in the graph. The color and size of the data point represent the direction and significance (*P*-value) of the correlation. Red indicates a positive correlation with drug resistance and green a positive correlation with drug sensitivity (i.e., proteins in red are more highly expressed in resistant cell lines).

B   AUC values for two Plk1 inhibitors are compared with expression of selected proteins (cMet, E-cadherin, β-catenin, and Snail); each data point represents an individual non–small-cell lung cancer cell line. The gray shaded area represents the 95% confidence interval around the linear regression (blue line). E-cadherin, β catenin, and cMet protein expression correlated with drug resistance.

but not in resistant/epithelial cells after treatment with volasertib (Fig 3C). However, changes in Src (Y416), FAK (Y925), and Akt (S473) were not statistically significant (Fig 3C). As with the RPPA results (Fig EV2A), the average decrease in FAK (Y397) phosphorylation was greater in the mesenchymal cells, but the effects of Plk1 inhibition on the epithelial cells were diverse (Fig 3C). Total cMet, FAK, and Src protein levels were not affected, supporting posttranslational changes. Likewise, when we knocked down Plk1 using siRNA, cMet (Y1234/1235) phosphorylation decreased only in sensitive/mesenchymal cells but not in resistant/epithelial cells with variable effects on pSrc and pFAK (Fig 3D).

Because cMet, FAK, and Src signaling can be bi-directional (Sen *et al*, 2011), we used inhibitors of cMet (tepotinib), FAK (VS6063), Src (dasatinib), and Plk1 (volasertib) to distinguish the properties of this signaling axis in NSCLC cell lines (Fig 3E). Consistent with our RPPA findings, these experiments showed that inhibition of Plk1 inhibits phosphorylation of cMet in the mesenchymal cell line but not in the epithelial one. Phosphorylation of the Plk1 substrate NPM1 was inhibited in both cell lines confirming that efficient target inhibition does not underlie the differential sensitivity. cMet inhibition also inhibited the phosphorylation of FAK in both cell lines, but FAK inhibition did not affect cMet activation. Inhibition of Plk1,

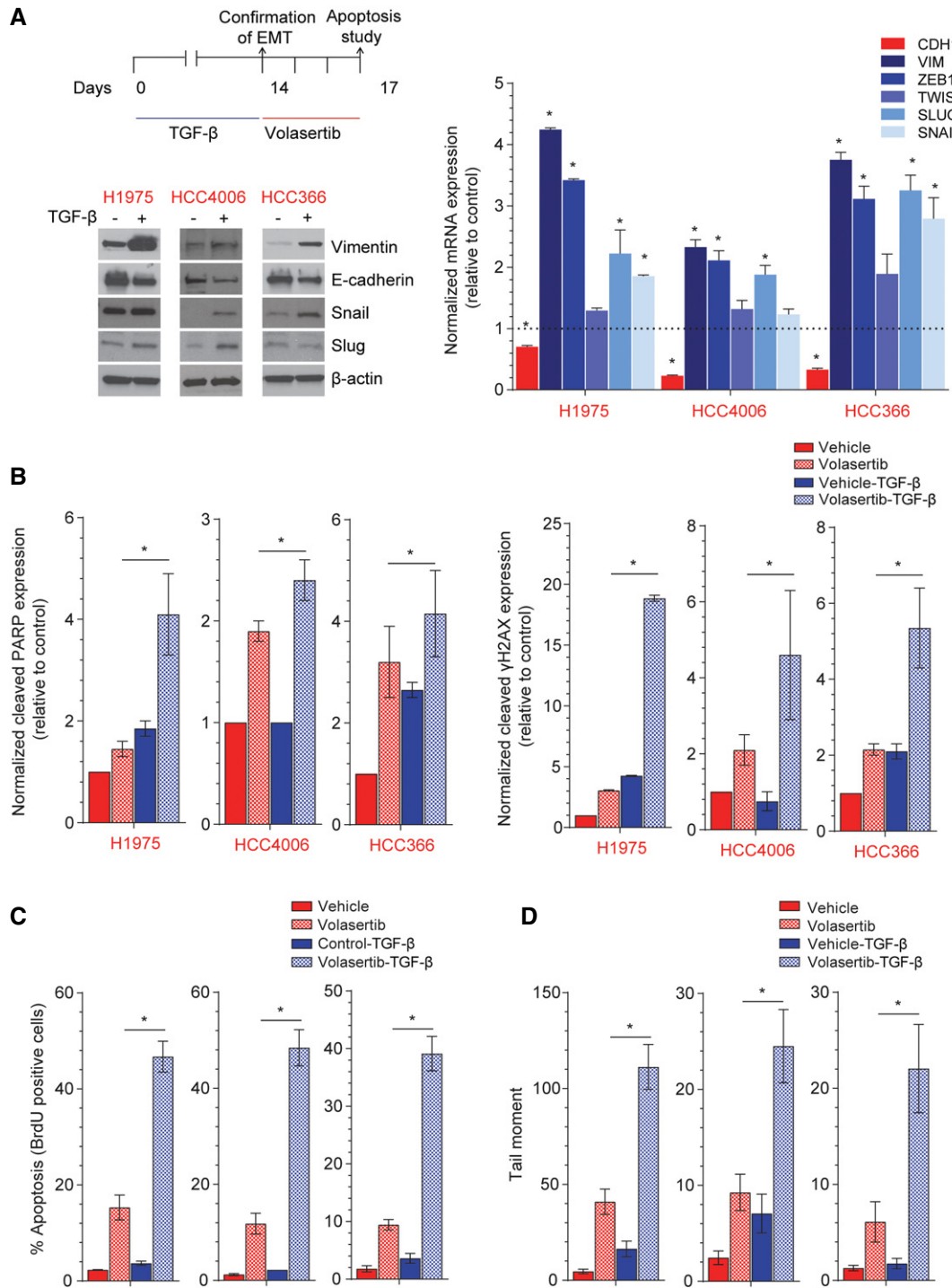

**Figure 2. TGF-β–induced mesenchymal phenotype increases sensitivity to Plk1 inhibition.**

A   Three epithelial/resistant non–small-cell lung cancer cell lines were treated with 5 ng/ml TGF-β for 14 days to induce a mesenchymal phenotype, which was confirmed with Western blot (left) and qPCR (right) analysis of epithelial-to-mesenchymal transition (EMT) markers.

B   Parental and TGF-β isogenic cell lines were treated with 50 nM volasertib for 72 h. Cells were then harvested, and lysates were immunoblotted for cleaved PARP and γH2AX proteins that were subsequently quantitated and normalized with β-actin.

C, D   Apoptosis was measured by the Apo-BrdU assay (C), and DNA damage was measured by the Comet assay (D).

Data information: Data are means ± standard error of the mean from three independent experiments. Significant differences using two-way analysis of variance with Bonferroni or Benjamini–Hochberg correction for multiple comparison are indicated (*$P < 0.01$).

Source data are available online for this figure.

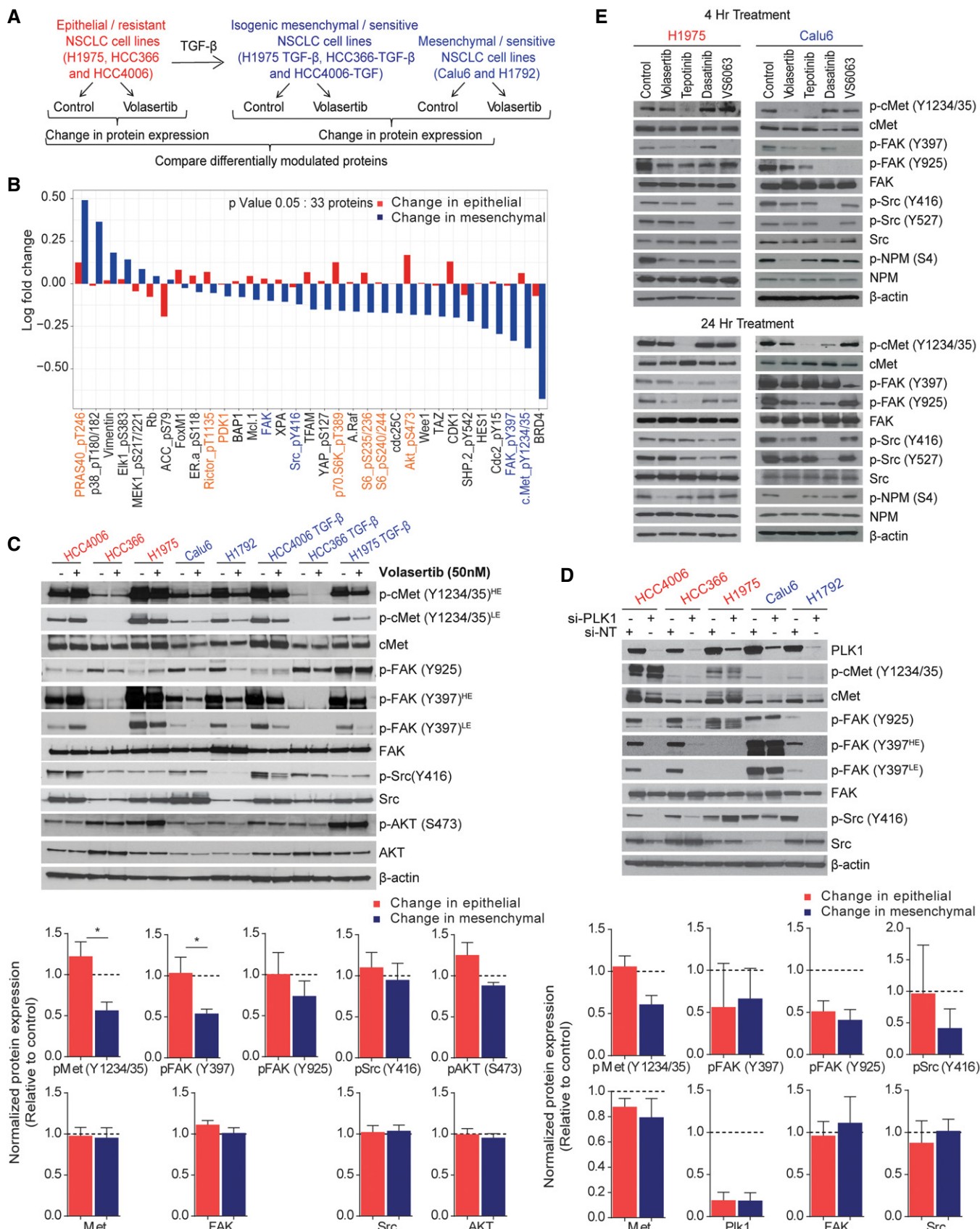

**Figure 3.**

**Figure 3. Activation of cMet is differentially regulated in epithelial and mesenchymal non–small-cell lung cancer (NSCLC) cell lines following Plk1 inhibition and knockdown.**

A  Three pairs of TGF-β–treated isogenic NSCLC cell lines (H1975, HCC4006, and HCC366) and two mesenchymal NSCLC cell lines (Calu6 and H1792) were incubated with 50 nM volasertib for 24 h, lysed, and subjected to reverse phase protein array analysis. Experiments were done in triplicate.

B  Thirty-three proteins were differentially regulated between epithelial and mesenchymal NSCLC cell lines after treatment with volasertib, including those involved in the cMet/FAK/Src signaling axis (blue text) and those involved in the PI3K/Akt signaling axis (orange text).

C  The same cell lines treated identically were subjected to immunoblotting for the indicated proteins (upper) with densitometric quantification normalized with β-actin (lower).

D  Epithelial (red text) and mesenchymal (blue text) NSCLC cell lines transfected with 10 nM siRNA as indicated for 48 h and subjected to Western blotting with the indicated antibodies (upper) with densitometric quantification normalized with β-actin (lower). NT, non-targeting control.

E  H1975 and Calu6 cells were treated with indicated inhibitors for 4 h or 24 h. Cells were then harvested, and lysates were immunoblotted for the indicated proteins.

Data information: Data are means ± standard error of the mean from three independent experiments. Significant differences using two-sided Student's t-test are indicated by *$P < 0.05$.

Source data are available online for this figure.

FAK, or cMet only minimally affected Src activation. Src inhibition decreased phosphorylation of FAK (Y925) but did not significantly affect phosphorylation of cMet or FAK (Y397). Together, these results support a model in which Plk1 is robustly inhibited in all NSCLC cell lines, but only inhibits cMet phosphorylation in mesenchymal NSCLC lines.

## The combination of Plk1 and cMet inhibition or knockdown enhances apoptosis in NSCLC cells

To test the hypothesis that cMet inhibition is important for Plk1 inhibitor–induced apoptosis, we treated eight NSCLC cell lines with a combination of volasertib and tepotinib 72 h and measured viability (Fig 4A). To inhibit cMet, we chose tepotinib, which has been tested in NSCLC patients (Friese-Hamim et al, 2017; Reungwetwattana et al, 2017). We used relevant drug concentrations as defined by pharmacokinetic data and target inhibition data. Specifically, 400 nM tepotinib fully inhibits cMet in intact NSCLC cells (Bladt et al, 2013). Results of in vitro kinase assays with 242 kinases showed that only cMet had half-maximal inhibitory concentration values of less than 600 nM (Bladt et al, 2013). None of the cell lines used in this research harbor MET mutations or MET amplification. A synergistic or additive effect was observed in seven of eight cell lines (Fa = 0.5; Fig 4B and Appendix Table S2). Likewise, the combination led to more apoptosis than did single-agent treatment in two epithelial and two mesenchymal cell lines, as measured by BrdU, cleaved PARP, and cleaved caspase 3 (Fig 4C and D). We also observed higher DNA damage (γ-H2AX expression) in all cell lines after treatment with the combination compared with single-agent treatment or controls (Fig 4D).

We also assessed colony formation in these cells after 24 h of treatment with vehicle control, volasertib, tepotinib, or the combination followed by drug-free incubation for 12–15 days. The combination treatment significantly decreased the number and size of colonies in all cell lines tested compared with control or single-agent treatment, although, consistent with our prior results, Plk1 inhibition alone was effective in mesenchymal NSCLC cell lines (Fig 4E and Appendix Fig S4).

To demonstrate the specificity of Plk1 inhibitors, we knocked down PLK1 and MET expression in NSCLC cell lines using siRNA for 48 h (Fig 4A) and observed a significant increase in apoptosis compared with non-targeting control and single-gene silencing (Fig 4F). Consistent with our inhibitor studies, silencing of Plk1 alone significantly increased the percentage of apoptotic cells in mesenchymal cell lines,

and we observed persistent cMet (Y1234/1235) phosphorylation in epithelial/resistant cell lines and decreased cMet activation in mesenchymal/sensitive cell lines (Fig 4G). All tested cell lines demonstrated significant increases in expression of cleaved PARP, cleaved caspase 3, and γ-H2AX in combination silencing compared with non-targeting control or single-gene silencing (Fig 4G). These results demonstrate that simultaneous inhibition or silencing of cMet potentiates the apoptotic effect of Plk1 inhibition or silencing in NSCLC.

## Inhibition of both Plk1 and cMet is more effective than inhibition of either target alone in vivo in NSCLC cell line and patient-derived xenograft (PDX) models

Encouraged by the in vitro activity, we next investigated the in vivo effect of Plk1 and cMet inhibition for the treatment of lung cancer in PDX and cell line xenograft models of NSCLC (Hao et al, 2015). We selected both epithelial (TC402) and mesenchymal (TC424) PDX and cell line models. We confirmed the xenografts' EMT status by checking the expression of EMT-related genes (Fig EV3A).

When tumors reached 150 mm³, mice were randomized into one of four treatment groups: vehicle control, volasertib, tepotinib, or the combination of volasertib and tepotinib. Volasertib alone resulted in a significant reduction in tumor growth compared with vehicle control or tepotinib alone in both models. As expected, volasertib reduced tumor size more significantly in the mesenchymal than in the epithelial models (Figs 5A and B, and EV3B). Specifically, volasertib alone led to tumor regression (i.e., tumor smaller than at the start of the experiment) in eight of 10 mesenchymal PDX mice, with an overall −26.5% regression in tumor volume relative to day zero (immediately prior to starting treatment). In contrast, volasertib alone led to tumor regression in only one of 10 epithelial PDX mice and overall tumor size increased by 156% compared with day zero (Figs 5B and EV3C). Likewise, in cell line xenografts (Fig EV3D), Calu6 (mesenchymal) cells showed a better response than did H1975 (epithelial) cells when treated with volasertib alone, although tumors did not regress in either model (Figs 5C and D, and EV3E).

The combination of Plk1 and cMet inhibitors led to tumor regression in both of the PDX models that was statistically significant starting on day 7 ($P < 0.05$). The combination therapy led to tumor regression in nine of 10 epithelial and nine of 10 mesenchymal PDX mice (Figs 5B and EV3C). Likewise, both cell line models showed a significant reduction in tumor growth when the mice were treated with the combination compared with control and with tepotinib alone

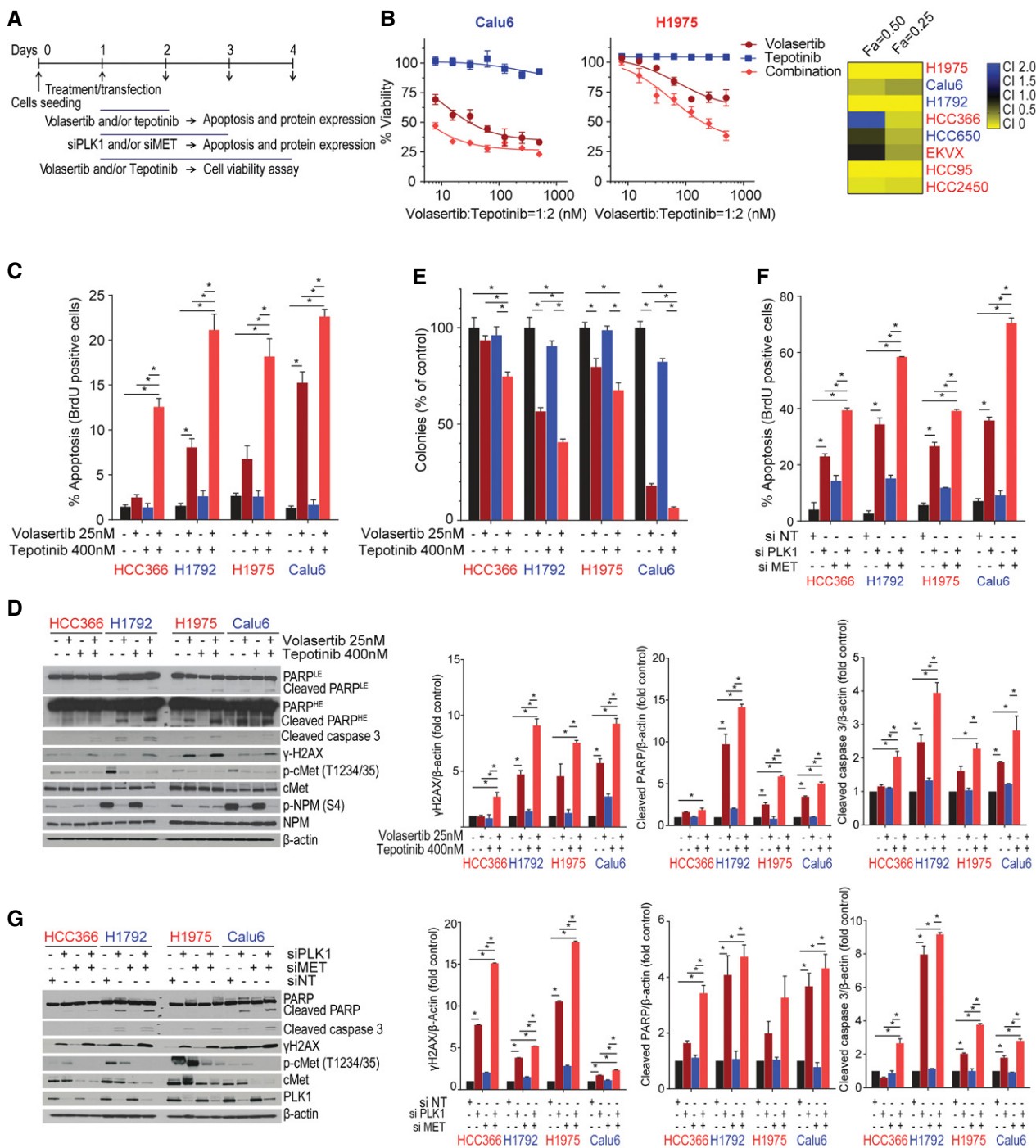

**Figure 4. Co-targeting of cMet and Plk1 enhances apoptosis in non–small-cell lung cancer (NSCLC) *in vitro*.**

A   Schematic of the experimental plan for Plk1 and cMet inhibition/silencing for viability and apoptosis studies in mesenchymal and epithelial NSCLC cell lines.

B   Cell viability was measured using CellTiter-Glo in NSCLC cell lines treated with volasertib, tepotinib, or a 1:2 ratio of both for 72 h. Left: representative cell viability graph of cells treated with the indicated drugs. Right: heatmap depicting the calculated combination indices at Fa = 0.25 and Fa = 0.5.

C   Apoptosis was measured using the Apo-BrdU assay in the indicated cell lines treated with the indicated drugs for 24 h.

D   Immunoblots from cells treated with indicated drugs for 24 h (left) with densitometric quantification normalized with β-actin (right).

E   All tested cell lines were treated as indicated for 24 h and allowed to grow in drug-free medium for 15–20 days to form colonies, which were counted using ImageJ.

F   Apoptosis was measured using the Apo-BrdU assay in the indicated cell lines transfected with 10 nM siRNA as indicated for 48 h. NT, non-targeting control.

G   Immunoblots from cells transfected with siRNA as indicated for 48 h (left) with densitometric quantification normalized with β-actin (right).

Data information: Data are means ± standard error of the mean from three independent experiments. Significant differences using two-way analysis of variance with Bonferroni or Benjamini–Hochberg (BH) correction for multiple comparison are indicated (*P < 0.01). Mesenchymal and epithelial NSCLC cell lines are indicated in blue and red text, respectively.

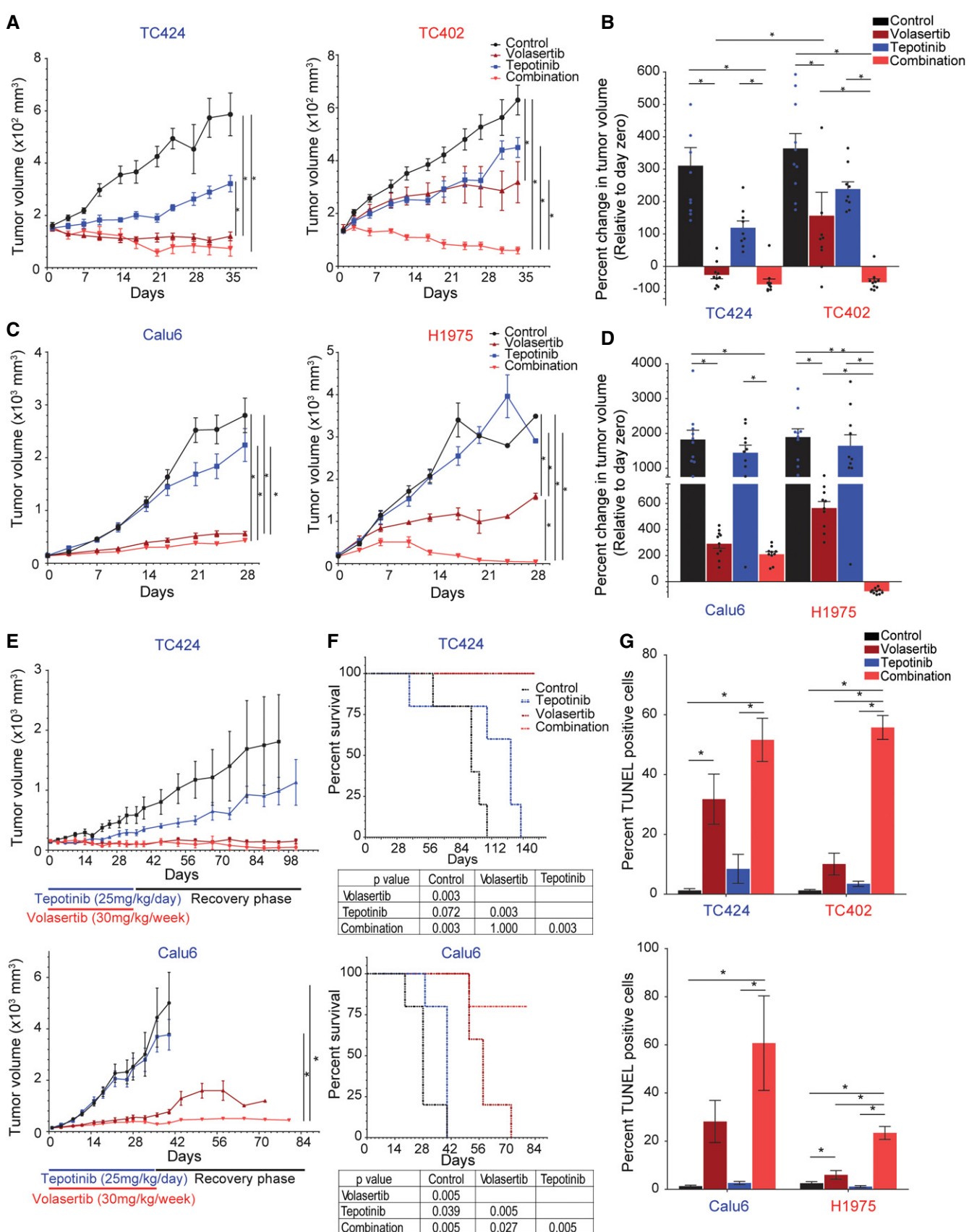

Figure 5.

**Figure 5. Co-targeting of cMet and Plk1 induces apoptosis and reduces tumor size in non–small-cell lung cancer.**

A–D Mice bearing epithelial (red text) and mesenchymal (blue text) non–small-cell lung cancer tumors were treated with vehicle control, volasertib (30 mg/kg per week intravenously), tepotinib (25 mg/kg per day orally), or the combination for 5 weeks to generate tumor growth curves of patient-derived xenografts (PDX; A) and cell line xenografts (C). The percent change in tumor volume at the end of treatment (normalized to day zero) was calculated with each data point representing an individual mouse, and the bar is the mean value ± standard error of 10 mice for each group (B, D).

E Tumor growth curve of mesenchymal PDX TC424 (top) and cell line xenograft Calu6 (bottom) upon treatment for 5 weeks and after cessation of treatment (recovery phase). Data are means ± standard error of the mean from five mice for each group.

F Kaplan–Meier survival curves for mesenchymal PDX TC424 (top) and cell line xenograft Calu6 (bottom) with death due to excessive tumor burden as the endpoint.

G Quantification of TUNEL-positive cells in paraffin-embedded xenograft tissue sections. Data are means ± standard error of the mean from six mice for each group.

Data information: Significant differences using two-way analysis of variance with Bonferroni or Benjamini–Hochberg correction for multiple comparisons are indicated.
*$P < 0.01$.
Source data are available online for this figure.

($P < 0.05$; Fig 5D). All 10 of the mice bearing the epithelial xenograft showed tumor regression (Figs 5D and EV3F). In the mesenchymal xenograft, the combination was slightly more effective than volasertib alone, but this difference did not reach statistical significance. In both cell line models, mice in the combination arm showed a significant improvement in survival compared with control; volasertib alone improved the survival of mice bearing mesenchymal NSCLC (Fig EV3G). The combination treatment was well tolerated in all mice with no change in body weight over time (Fig EV3H).

To investigate the ability of treated tumors to recover, we treated mice bearing mesenchymal NSCLC for 5 weeks, then stopped drug treatment and examined the mice for tumor growth. The tumors in the mice treated with vehicle control or tepotinib continued to grow steadily. In contrast, tumors remained the same size in both the volasertib-only and combination treatment groups, with the exception of modest tumor growth in the volasertib-only Calu6-bearing mice (Fig 5E). The volasertib-only and combination treatment groups had significantly longer survival compared with other groups in both models ($P < 0.01$; Fig 5F).

To asses for apoptosis, we next performed TUNEL staining in paraffin-embedded tissues (Fig 5G). Similar to the *in vitro* finding, volasertib alone resulted in a larger increase in TUNEL-positive cells in the mesenchymal xenograft models (TC424 and Calu6) than in the epithelial xenograft models (TC202 and H1975). Co-treatment with Plk1 and cMet inhibitors significantly increased the percentage of TUNEL-positive cells in all mouse models. Taken together, these results support cMet as a driver of Plk1 inhibitor resistance in epithelial NSCLC *in vivo*, and the findings suggest that co-inhibition of Plk1 and cMet increases apoptosis, leading to tumor regression in NSCLC.

### Constitutive activation of cMet abrogates Plk1 inhibition–induced apoptosis in mesenchymal NSCLC cell lines

To test cMet as a driver of resistance, we used constitutively active TPR-Met, which is a fusion protein that lacks the extracellular, transmembrane, and juxtamembrane domains of the cMet receptor and includes the TPR dimerization motif, which allows constitutive and ligand-independent activation of the kinase. Expression of the TPR-Met chimeric protein was biologically active in three mesenchymal cell lines (Calu6, H157, and H1355), which in turn increased downstream Akt (S473) and Erk (p42/44 T202/204) phosphorylation (Fig EV4A).

Compared with the control vector–transfected cells, TPR-Met–expressing cells showed less apoptosis, as measured by Apo-BrdU, cleaved PARP, and cleaved caspase 3, after treatment with

volasertib (Fig 6A and B). Likewise, expression of TPR-Met increased resistance to Plk1 inhibition compared with control vector, as measured by colony formation (Figs 6C and EV4B) assays and CellTiter-Glo (Fig EV4C).

To determine whether we could overcome the effects of cMet activation by using a cMet inhibitor, we combined tepotinib and volasertib. Tepotinib inhibited cMet universally but did not completely abrogate cMet signaling in the TPR-Met–expressing cell lines. The addition of tepotinib significantly increased apoptosis in control vector–transfected cells, but apoptosis was only moderately increased in TPR-Met–transfected cells, consistent with a lack of full cMet inhibition (Fig 6A and B). The combination of volasertib and tepotinib significantly reduced the clonogenic potential compared with single-agent treatment in both control and TPR-Met–transfected cell lines (Fig 6C). Taken together, these results support cMet as a driver of Plk1 inhibitor sensitivity in mesenchymal NSCLC.

The ability of constitutively active TPR-Met to mediate resistance to Plk1 inhibition led us to ask whether Plk1 inhibition would affect cMet activation in NSCLC with constitutively active cMet. To test this hypothesis, we inhibited Plk1 in mesenchymal NSCLC cell lines with *MET* amplification and a *MET* exon 14 skipping mutation (Fig 6D). In both the cell line with marked *MET* amplification (H920, copy number 16.4) and the one with an activating *MET* mutation (H596), volasertib did not lead to a decreased in pMet. Both H596 and H920 were resistant to the Plk1 inhibition GSK461364 (Ferrarotto *et al*, 2016).

### Acquired resistance to volasertib leads to a mesenchymal-to-epithelial transition and decreased Plk1 inhibitor–mediated cMet inhibition

To study the molecular mechanism by which acquired resistance to volasertib emerges, we exposed the Calu6 cell line to increasing concentrations of the drug over time until resistance emerged (Fig 7A). The resulting Calu6-VAR (volasertib acquired resistance) cell line showed an epithelial morphology (Fig 7B), increased expression of E-cadherin, and decreased expression of vimentin compared with the parental cell line. Similar to the protein changes, mRNA expression also showed about a 70-fold increase in *CDH1* ($P < 0.01$) and a decrease in the mesenchymal genes *VIM* (0.1-fold), *ZEB1* (0.001-fold), *TWIST* (0.3-fold), and *SNAIL* (0.001-fold; $P < 0.01$) compared with the parental cell line (Fig 7C). We did not observe any significant change in basal cMet or p-cMet (Y1234/1235) protein expression in Calu6-VAR cells compared with parental cells. We observed persistent p-cMet (Y1234/1235) phosphorylation

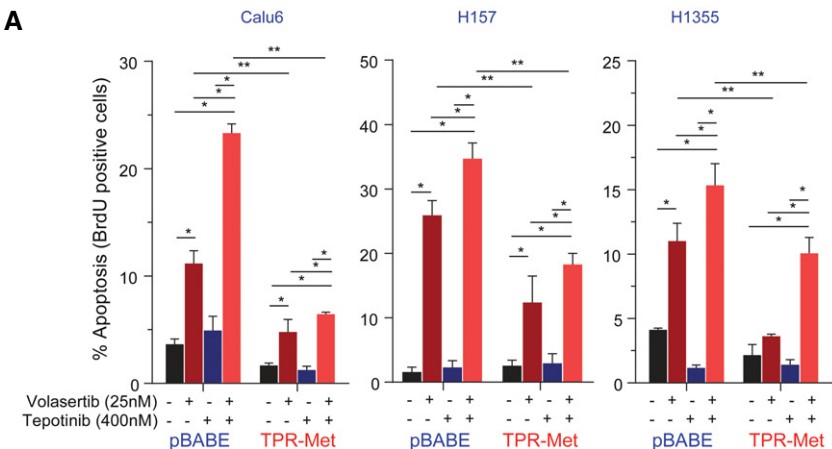

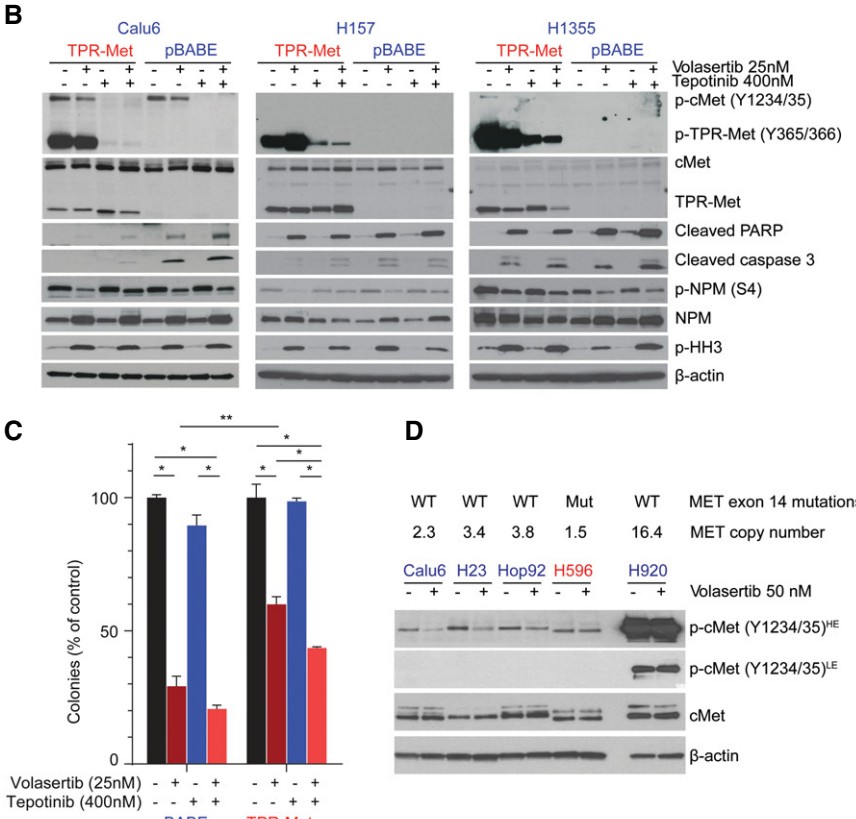

**Figure 6.  Constitutively active cMet expression reduces sensitivity to Plk1 inhibition.**

A   Calu6, H157, and H1355 TPR-Met–expressing or vector control (pBABE)–expressing cells were treated as indicated for 24 h, and apoptosis was measured using the Apo-BrdU assay.

B   Parental and TPR-Met–expressing cell lines were treated with the indicated drugs for 24 h. Cells were then harvested, and lysates were immunoblotted for the indicated proteins. β-Actin was used as a loading control.

C   Parental and TPR-Met–expressing cell lines were treated as indicated for 24 h and allowed to grow in drug-free medium for 15–20 days to form colonies, which were counted using ImageJ.

D   Mesenchymal NSCLC cell lines with the noted MET alterations were incubated with 50 nM volasertib for 4 h and subjected to immunoblotting with the indicated antibodies.

Data information: (A and C) Data are means ± standard error of the mean from three independent experiments. Significant differences using two-way analysis of variance with Bonferroni or Benjamini–Hochberg correction for multiple comparisons are indicated. *P < 0.05. **P < 0.01.
Source data are available online for this figure.

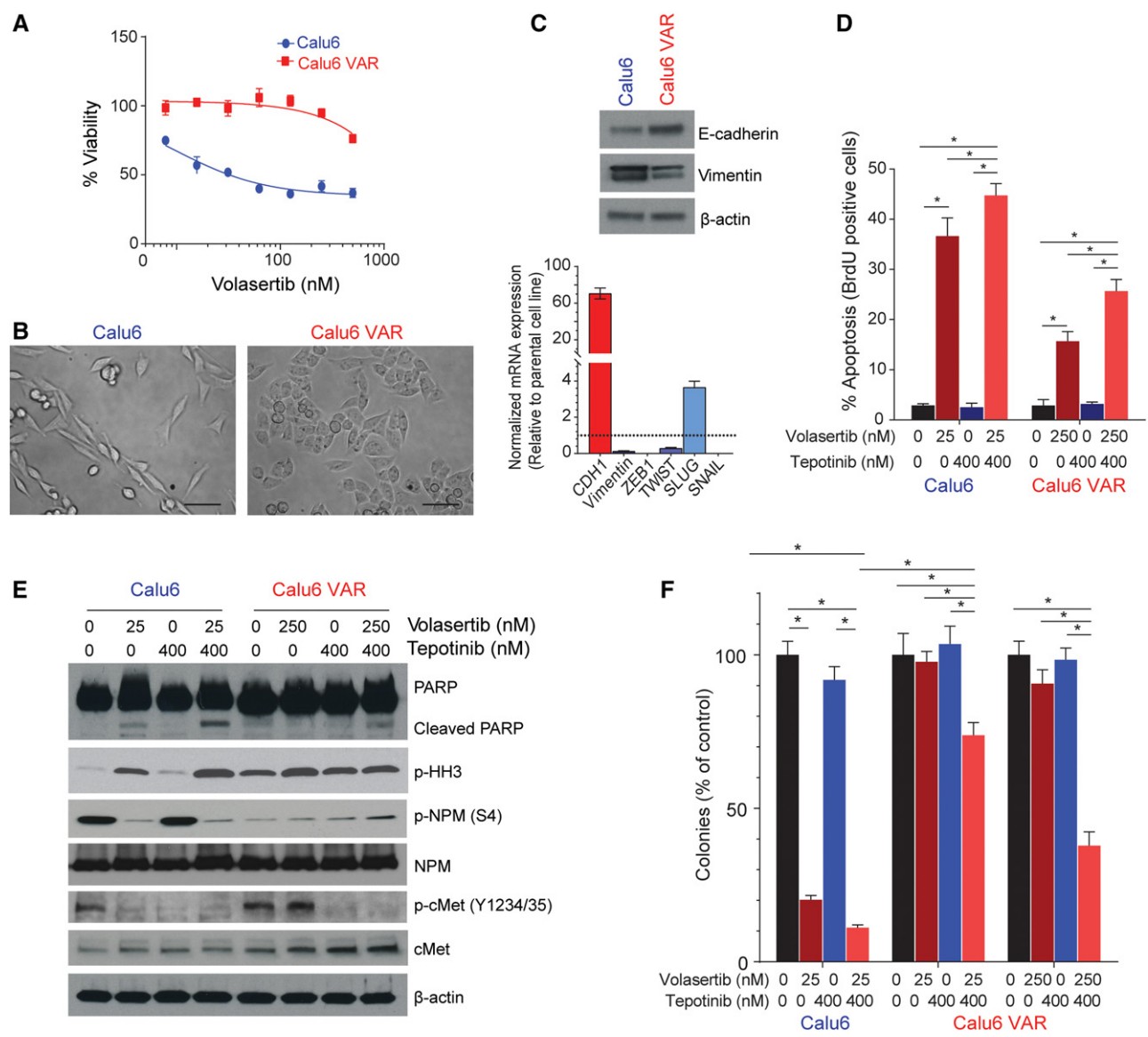

**Figure 7. The volasertib acquired resistance (VAR) cell line shows mesenchymal-to-epithelial transition and persistent cMet phosphorylation.**

A Cell viability measured by CellTiter-Glo in Calu6 parental and Calu6-VAR cells treated with different concentrations of volasertib for 72 h.

B Micrograph shows acquisition of a mesenchymal-to-epithelial transition phenotype in VAR cells. Scale bar: 100 μm.

C Basal protein and mRNA expression in parental and VAR cells for the indicated proteins was measured using immunoblotting (upper) and reverse-transcription PCR (lower), with GAPDH expression used for normalization.

D Parental and VAR cells were treated as indicated for 48 h, and apoptosis was measured using the Apo-BrdU assay.

E Parental and VAR cells were treated with the indicated drug concentrations for 48 h. Cells were then harvested, and lysates were immunoblotted for the indicated proteins.

F Parental and VAR cells were treated as indicated for 24 h and allowed to grow in drug-free medium for 15–20 days to form colonies, which were counted using ImageJ.

Data information: Data are means ± standard error of the mean from three independent experiments. Significant differences using two-way analysis of variance with Bonferroni or Benjamini–Hochberg correction for multiple comparisons are indicated. *$P < 0.01$.

Source data are available online for this figure.

---

after treatment with volasertib in Calu6-VAR cells, as well as in cell lines with *de novo* resistance.

The differential effects on cMet activation led us to analyze the effect of co-targeting Plk1 and cMet with volasertib and tepotinib on apoptosis and cell viability. Plk1 inhibition alone led to less

apoptosis in Calu6-VAR cells than in parental cells ($P < 0.01$), as measured by Apo-BrdU and cleaved PARP (Fig 7D and E). Simultaneous inhibition of both cMet and Plk1 significantly increased the percentage of apoptotic BrdU-positive cells and PARP cleavage compared with single-agent treatment ($P < 0.05$) in both parental

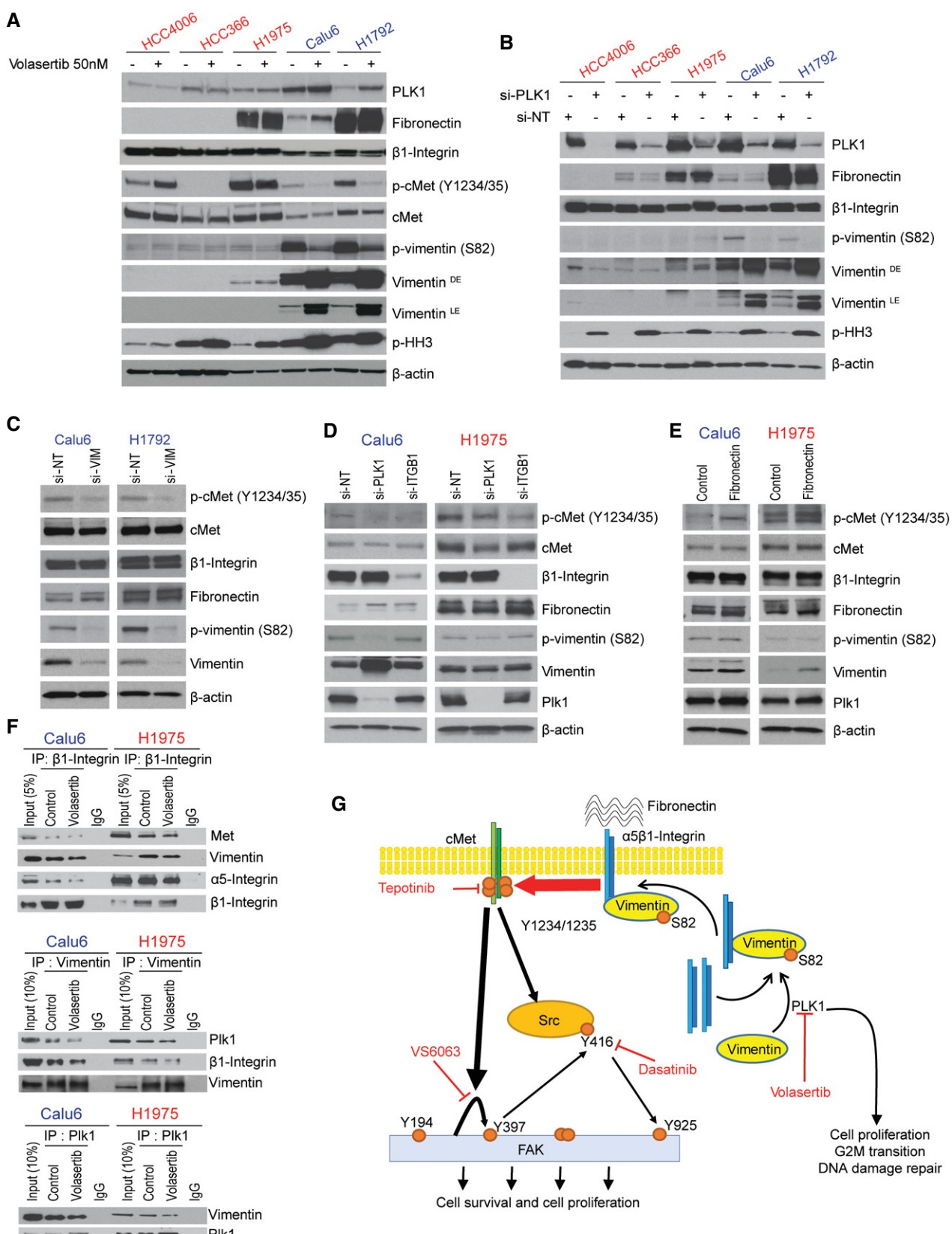

**Figure 8.**

**Figure 8.   Plk1 inhibition or silencing decreases vimentin phosphorylation and regulates cMet phosphorylation via β1-integrin.**

A   Epithelial and mesenchymal non–small-cell lung cancer (NSCLC) cell lines after treatment with 50 nM volasertib for 24 h. Cells were then harvested, and lysates were immunoblotted for the indicated proteins.
B   Epithelial and mesenchymal NSCLC cell lines after Plk1 silencing with siRNA for 48 h. Cells were then harvested, and lysates were immunoblotted for the indicated proteins.
C   Calu6 and H1792 mesenchymal NSCLC cell lines after *VIM* silencing with siRNA for 48 h. Cells were then harvested, and lysates were immunoblotted for the indicated proteins.
D   Calu6 (mesenchymal) and H1975 (epithelial) NSCLC cell lines after *PLK1* and *ITGB1* silencing with siRNA for 48 h. Cells were then harvested, and lysates were immunoblotted for the indicated proteins.
E   Calu6 (mesenchymal) and H1975 (epithelial) NSCLC cell lines after treatment with fibronectin for 24 h. Cells were then harvested, and lysates were immunoblotted for the indicated proteins.
F   Calu6 and H1975 NSCLC cell lines after treatment with 50 nM volasertib for 24 h. Cells were then harvested, and lysates were immunoprecipitated (IP) using β1-integrin, vimentin, and Plk1 antibodies, then immunoblotted for the indicated proteins.
G   Schematic summary of the proposed mechanism by which cMet drives primary and acquired resistance to volasertib in NSCLC and possible treatment combinations to overcome this resistance.

Source data are available online for this figure.

and Calu6-VAR cells (Fig 7D and E). We also assessed the colony formation of these cells after 24 h of treatment with vehicle control, volasertib, tepotinib, or both followed by drug-free incubation for 12–15 days. Volasertib alone was not effective in Calu6-VAR cells, but combination treatment significantly decreased the number of colonies with both low (25 nM) and high (250 nM) concentrations of volasertib ($P < 0.01$; Fig 7F and Appendix Fig S5A). Likewise, CellTiter-Glo analysis showed more sensitivity when Calu6 parental and Calu6-VAR cells were treated with the combination of volasertib and tepotinib compared with single-agent treatment (Appendix Fig S5B).

**Plk1 inhibition or silencing decreases vimentin phosphorylation and regulates cMet phosphorylation via β1-integrin trafficking**

To determine whether cMet inhibition following Plk1 inhibition is ligand-dependent, we measured hepatocyte growth factor (HGF) in conditioned medium and in cellular lysates after Plk1 inhibition in both mesenchymal and epithelial cell lines. Neither the secreted nor cellular HGF levels were affected by Plk1 inhibition in all tested NSCLC cell lines (Fig EV5A). Likewise, an HGF neutralizing antibody did not affect basal cMet activation or Plk1 inhibition–induced cMet inhibition in the mesenchymal NSCLC cell line Calu6. In a second mesenchymal NSCLC cell line (H1792), the HGF neutralizing antibody did reduce basal cMet activation consistent with cancer cell production of HGF leading to cMet activation that has previously been well established in some NSCLC tumors (Salgia, 2017). In H1792 cells, the combination of the HGF neutralizing antibody and Plk1 inhibition had an additive effect on cMet activation (Fig EV5B).

We next investigated the interaction of cMet, Plk1, β1-integrin (*ITGB1*), and vimentin (*VIM*) because this is a ligand-independent pathway that involves proteins associated with EMT. Plk1 maintains cell surface levels of β1-integrin via phosphorylation of vimentin on S82 (Rizki *et al*, 2007). β1-integrin can form a heterodimer complex with cMet, resulting in ligand-independent cross-activation invasive oncologic processes (Jahangiri *et al*, 2017). We hypothesized that Plk1 regulates vimentin phosphorylation, which in turn regulates the β1-integrin trafficking to the membrane and maintains cMet phosphorylation in a ligand-independent manner. Because mesenchymal cells have higher vimentin levels than epithelial cells do, cMet activation using this pathway would be more significant in

mesenchymal cells than in epithelial cells. To test this hypothesis, we first checked the expression of vimentin and β1-integrin in NSCLC cell lines. As expected, we observed a significant positive correlation between expression of mesenchymal markers, according to an EMT score (Byers *et al*, 2013), and expression of *ITGB1* ($r = 0.528$, $P < 0.001$) and *VIM* ($r = 0.777$, $P < 0.001$; Fig EV5C).

We then tested the effect of Plk1 inhibition or silencing on vimentin (S82) phosphorylation. Inhibition or silencing of Plk1 by volasertib or siRNA markedly decreased the S82 vimentin phosphorylation in two mesenchymal cell lines. The resulting increase in total vimentin may be a compensatory change. Vimentin levels were very low in epithelial cell lines without consistent effects by Plk1 inhibition (Fig 8A and B); total β1-integrin levels were not affected.

To test the role of vimentin, we silenced it in two mesenchymal cell lines, which led to a decrease in cMet phosphorylation at Y1234/1235 (Fig 8C), supporting the role of vimentin in cMet activation. We also observed an increase in cleaved PARP and caspase 3 expression when silencing of vimentin was followed by treatment with volasertib in these cell lines compared with cells treated with volasertib alone (Fig EV5D). Knockdown of vimentin or treatment with volasertib did not affect the expression of total levels of β1-integrin, cMet, or fibronectin (Figs 8C and EV5D).

To confirm the involvement of β1-integrin in cMet phosphorylation, we manipulated β1-integrin levels by siRNA and β1-integrin activation with fibronectin. Knocking down *ITGB1* in both epithelial and mesenchymal cell lines led to a decrease in cMet phosphorylation (Figs 8D and EV5E), demonstrating that this axis is intact in both cell types. Co-silencing of *Plk1* and *ITGB1* led to an increase in cleaved PARP and cleaved caspase 3 expression compared with *Plk1* silencing alone (Fig EV5E). Consistent with these results, treatment with fibronectin led to an increase in cMet phosphorylation in both epithelial and mesenchymal NSCLC cell lines (Fig 8E). Fibronectin was not able to abrogate Plk1 inhibitor–induced cMet inactivation and likewise had no effect in rescuing the cells from apoptosis by volasertib (Fig EV5F), likely because of a lack of available cell surface β1-integrin. Manipulation of β1-integrin did not affect the total levels of cMet, Plk1, or vimentin.

Next, we investigated the interaction between Plk1, vimentin, β1-integrin, and cMet in NSCLC cell lines after treatment with volasertib. Plk1, vimentin, and β1-integrin were immunoprecipitated from volasertib-treated and volasertib-untreated Calu6 and H1975 cell lysates, and immunoprecipitated protein was resolved by SDS–

PAGE and probed for Plk1, vimentin, β1-integrin, and cMet. Results in Fig 8F show that vimentin co-immunoprecipitated with Plk1. A reverse immunoprecipitation was performed, and Plk1 was found to co-immunoprecipitate with vimentin in both NSCLC lines. Similarly, β1-integrin co-immunoprecipitated with vimentin and vimentin co-immunoprecipitated with β1-integrin. We also observed that cMet co-immunoprecipitated with β1-integrin. Altogether, these findings confirm that cMet phosphorylation is regulated by Plk1-mediated vimentin phosphorylation via β1-integrin in these cells (Fig 8G).

## Discussion

The current study yielded five main findings that follow from our prior discovery that mesenchymal NSCLC is more sensitive to Plk1 inhibitors than epithelial NSCLC (Ferrarotto et al, 2016). First, Plk1 inhibitor sensitivity correlates with cMet and EMT protein expression in a large, independent dataset, validating our prior study. Second, cMet phosphorylation is differentially regulated after Plk1 inhibition in epithelial and mesenchymal NSCLC. Third, pharmacologic and genetic manipulation of cMet affects Plk1 inhibitor–induced apoptosis, establishing this pathway as a bona fide mechanism of resistance. This conclusion was further supported by the finding that a cell line with acquired resistance to Plk1 inhibition showed mesenchymal-to-epithelial transition and persistent cMet phosphorylation after Plk1 inhibition, similar to cells with de novo resistance. Fourth, this resistance pathway functions in vivo; significant tumor regression was demonstrated in multiple NSCLC models treated with clinically relevant drugs. Finally, the sensitivity of mesenchymal NSCLC cells to Plk1 inhibition is linked to cMet phosphorylation via the vimentin and β1-integrin pathway (a ligand-independent and understudied pathway), leading to cMet activation in NSCLC.

Our work has important clinical implications for the treatment of NSCLC. Previously, we demonstrated that 63 NSCLC cell lines have diverse sensitivities to Plk1 inhibition (Ferrarotto et al, 2016), which is consistent with the results of clinical trials of Plk1 inhibitors for solid tumors, in which response rates were low (4–14%) in unselected patients with stable disease rates of 26–42% (Schoffski et al, 2012, 2010; Sebastian et al, 2010; Stadler et al, 2014). Although approximately 20% of NSCLC tumors are mesenchymal (Akbani et al, 2014; Chen et al, 2014) and predicted to be sensitive to Plk1 inhibition, the reversibility of EMT and intratumoral heterogeneity may diminish the efficacy of Plk1 inhibitors, contributing to low response rates. Recently, Lee et al (2018) measured the same 76-gene EMT score that we had used to classify NSCLC cell lines (Byers et al, 2013) in 35 regions from 10 NSCLC tumors. Considerable differences in the EMT scores were observed between regions in two of the 10 tumors; in the other eight tumors, different regions within each tumor had very similar EMT scores. The finding that the combination of Plk1 and cMet inhibition led to striking tumor regression in vivo in both epithelial and mesenchymal NSCLC suggests that this combination would be broadly effective in NSCLC patients who may have tumors with heterogeneity or dynamic changes in EMT status. In addition, it is rational to target a pathway that may mediate acquired resistance upfront to achieve a more durable response to therapy (Neel & Bivona, 2017). Because both cMet and Plk1

inhibitors are in clinical development, our work could be rapidly translated to clinical testing.

Recently published work using a novel drug (Poloppin) that inhibits phosphopeptide binding by the Plk1 polo-binding domain supports our findings (Narvaez et al, 2017). Poloppin led to cell death in KRAS mutant cancer cells in vitro and decreased tumor size in vivo. Because cMet was upregulated in a colon cancer cell line (SW48) expressing $KRAS^{G12D}$, Narvaez, et al investigated the combination of MET knockdown and cMet inhibition with Poloppin. Both cMet inhibition and knockdown did sensitize five KRAS mutant cell lines and one pancreatic organoid to Poloppin in vitro.

Dysregulation of cMet signaling–mediated proliferation, apoptosis, and migration through cMet overexpression, MET amplification, MET mutation, or HGF-induced activation has been widely demonstrated in oncogenic processes across multiple tumor types, including NSCLC (Smyth et al, 2014; Tsuta et al, 2012; Van Der Steen et al, 2015). cMet inhibitors have been extensively studied in NSCLC (Salgia, 2017). Recently, mutations in exon 14 of MET that lead to an in-frame deletion of the negative regulatory juxtamembrane domain were identified in 4% of lung adenocarcinomas. These mutations result in cMet activation and clinical sensitivity to cMet inhibition (Paik et al, 2015). cMet inhibitors are actively being studied in cancers bearing these mutations; a phase II clinical trial of tepotinib in NSCLC harboring MET exon 14 skipping alterations (NCT02864992) is underway. MET is also amplified in 5% of newly diagnosed lung adenocarcinomas (Cappuzzo et al, 2009; Kong-Beltran et al, 2006; Tsuta et al, 2012), and cMet inhibitors are effective in patients with gene copy numbers > 5 (Salgia, 2017). The presence of cMet/GRB2 complexes is associated with response to cMet inhibitors and may serve as an additional biomarker to select patients who are sensitive to cMet inhibitors in the future (Smith et al, 2017). In the current study, the expression of total cMet protein predicted resistance to four Plk1 inhibitors, although cMet protein and gene expression did not correlate with response in our prior study (Ferrarotto et al, 2016). Our data suggest that vimentin-dependent cMet activation is independent of canonical mechanisms of cMet activation because a HGF neutralizing antibody can still affect cMet activation in mesenchymal NSCLC and because robust cMet activation by an activating mutation or amplification was not affected by Plk1 inhibition.

Several reports established the cooperativity of cMet and the epidermal growth factor receptor (EGFR) in various cancers, including NSCLC. Co-targeting of these kinases potentially produces synergistic antitumor effects (Chae et al, 2016; Puri & Salgia, 2008; Wu et al, 2016a), although responses in clinical studies of the combination in NSCLC patients have been modest (Ma, 2015). Both cMet activation and EMT function as resistance mechanisms to EGFR tyrosine kinase inhibitors in NSCLC (Jakobsen et al, 2017; Ninomiya et al, 2018; Rotow & Bivona, 2017; Stewart et al, 2015; Yoshida et al, 2016). Clinical studies combining cMet and EGFR inhibitors in MET-amplified, EGFR-mutant NSCLC are ongoing (Rotow & Bivona, 2017). We previously tested the efficacy of Plk1 inhibition in NSCLC cell lines with acquired resistance to EGFR inhibitors and found that cell lines that had undergone EMT, but not those with T790M mutations, became more sensitive to Plk1 inhibition (Wang et al, 2016).

During the reversible process of EMT, epithelial cells lose cell–cell contacts, increase motility, and develop resistance to apoptosis

induced by chemotherapy, targeted therapy, and radiotherapy. Changes in gene expression during EMT reflect these characteristics and include repression of E-cadherin expression with the dissolution of cell–cell junctions; changes in genes encoding cytoskeletal proteins, including the activation of vimentin; and alterations in proteins that affect extracellular matrix interactions, including β1-integrin. Inhibition of EMT in genetically diverse NSCLC cell lines led to growth inhibition (Burns *et al*, 2013). Recent studies have demonstrated that Plk1 has a cell cycle–independent function in the regulation of EMT by activating CRAF/ERK signaling in prostate cancer (Wu *et al*, 2016b) or by activating AKT in gastric cancer (Cai *et al*, 2016). Plk1 may also indirectly contribute to EMT through regulation of its substrate, the transcription factor Forkhead box M1 (Kong *et al*, 2014; Wang *et al*, 2014). However, when we manipulated Plk1, NSCLC cells did not undergo morphologic changes consistent with EMT or show changes in expression of EMT-related proteins, suggesting that Plk1 is not a driver of EMT in NSCLC (Ferrarotto *et al*, 2016).

The link between EMT and Plk1 inhibition–induced apoptosis is not explained by canonical signaling pathways, but we hypothesized that proteins that were differentially expressed during EMT, such as vimentin, might be important in this process. CDK1 phosphorylates vimentin at S56, which provides a Plk1 binding site. Plk1 further phosphorylates vimentin at S82 (Oguri *et al*, 2006; Yamaguchi *et al*, 2005), which regulates vimentin filament segregation, coordinately with rho-kinase and Aurora-B (Yamaguchi *et al*, 2005). Recently, another study also demonstrated that Plk1 regulates smooth muscle contraction by modulating vimentin phosphorylation at S56 (Li *et al*, 2016). In addition, Plk1 regulates cell surface levels of β1-integrin and invasion by phosphorylating vimentin in breast cancer cells (Rizki *et al*, 2007).

Integrins and other cell surface receptors can lead to ligand-independent activation of cMet (Mitra *et al*, 2011; Varkaris *et al*, 2013). β1-Integrin positively regulates the endocytosis of activated cMet, as well as cMet signaling after endocytosis in some breast and lung cancer cell lines. cMet and β1-integrin form a complex on the plasma membrane. cMet activation in either a ligand-dependent or ligand-independent manner results in internalization and activation of β1-integrin. Additionally, β1-integrin in its active conformation positively regulates the endocytosis of activated cMet. Thus, both cMet and β1-integrin mutually interact with each other for optimal internalization in a clathrin-dependent manner. After internalization, the cMet/β1-integrin complex gradually accumulates on autophagy-related endomembranes, where β1-integrin promotes downstream cMet signaling via p52Shc adaptor protein that in turn activates the ERK1/2 signaling pathway. This β1-integrin–dependent cMet-sustained signaling regulates cell survival and growth, tumorigenesis, invasion, and lung colonization *in vivo* (Barrow-McGee *et al*, 2016).

Recently, Jahangiri *et al* demonstrated cMet displaces the α-heterodimer partner of β1-integrin, resulting in cMet/β1-integrin complex formation. This switching of the β1-integrin binding partner from α5-integrin to cMet significantly increases fibronectin affinity. Because β1-integrin lacks enzymatic activity, its signaling depends on the kinase adaptor protein ILK, which binds the cytoplasmic domain of β1-integrin. In the cMet/β1-integrin complex, fibronectin promotes ILK-mediated cMet phosphorylation even in the absence of HGF. Formation of this complex is regulated by tumor microenvironmental factors associated with metastasis and therapeutic resistance (Jahangiri *et al*,

2017). This activation could lead to Plk1 inhibitor resistance in mesenchymal NSCLC by bypassing Plk1 inhibition–mediated cMet inhibition. This residual cMet/β1-integrin complex activity, which is dependent upon ligands in the tumor microenvironment, may explain why the combination of PLK1 and cMet inhibition is more effective than single drugs in some mesenchymal NSCLC models and why we observed differences in single-agent PLK1 inhibitor efficacy *in vitro* compared with *in vivo*. Consistent with that study, we also observed that cMet and β1-integrin interact in NSCLC and that manipulation of β1-integrin affected cMet phosphorylation. Furthermore, we demonstrated that Plk1 regulates vimentin phosphorylation, which in turn regulates cMet phosphorylation. We are the first to connect these signaling pathways from Plk1 to cMet and the first to study this pathway in NSCLC. Future studies will include fibronectin affinity studies and subcellular localization studies of β1-integrin, cMet, vimentin, and Plk1 to assess the specific effects of these pathways on β1-integrin trafficking.

In conclusion, we demonstrated that Plk1 inhibition leads to apoptosis in mesenchymal NSCLC owing to direct effects on Plk1, as well as parallel vimentin and β1-integrin–mediated, ligand-independent inhibition of cMet. Moreover, the lack of cMet inhibition in epithelial NSCLC defines a previously unknown pathway of resistance to Plk1 inhibition. Although EMT proteins could serve as candidate biomarkers to select patients for Plk1 inhibition, intratumoral heterogeneity of EMT may limit this approach. The addition of cMet inhibitors is a promising therapeutic strategy to overcome *de novo* and acquired resistance to Plk1 inhibitors in patients with NSCLC.

## Materials and Methods

### Reagents and cell lines

All drugs were purchased from Selleck Chemicals (Houston, TX) and prepared as 10 mmol/l stock solutions in dimethyl sulfoxide. Predesigned sets of four independent siRNA sequences of the target genes *Plk1* and *MET* (siGENOME SMARTpool; Dharmacon, Lafayette, CO) were used. Human TGF-β1 was purchased from Cell Signaling Technology (Danvers, MA). Human fibronectin was purchased from Sigma-Aldrich (St. Louis, MO). Constitutive active TPR-Met plasmid and control pBABE plasmid were obtained from Addgene (Cambridge, MA). Antibodies and dilutions used in the study are listed in Appendix Table S3. The HGF neutralizing monoclonal antibody (24612.111) was acquired from Invitrogen (Catalog # MA1-24767, Thermo Fisher Scientific, Waltham, MA).

Human NSCLC cell lines were obtained, maintained, and genotyped by STR profiling as previously described (Ferrarotto *et al*, 2016; Peng *et al*, 2016) and routinely tested for the presence of mycoplasma species using the Mycoplasma Detection Kit (Lonza, Basel, Switzerland).

### Development of VAR cell line

For generation of the VAR cell line, the Calu6 cell line was incubated with stepwise increasing concentrations of volasertib. The established Calu6 VAR cell line was maintained in 250 nM volasertib.

### Transfections and transduction

Retroviruses were produced by co-transfecting HEK293 cells with 1 μg of the viral packaging vector pUMVC3 and envelope vector pCMV-VSV-G (8:1 ratio) and 1 μg of retroviral plasmid pBABE–TPR-Met or the control vector using Lipofectamine 3000. The HEK293 cell medium was changed 24 h after transfection, and the cells were incubated at 37°C for 48 h to allow for virus production. After 48 h, HEK293 medium containing viral particles was filtered and transferred onto NSCLC cell culture plates and incubated at 37°C for 48 h. After transduction, fresh RPMI 1640 medium with 10% fetal bovine serum was added to the cell culture plates, and the NSCLC cells were allowed to recover for 24 h. NSCLC cells were selected using 3 μg/ml puromycin.

For knockdown experiments, NSCLC cells were transfected with predesigned sets of four independent siRNAs against Plk1 and cMet using Lipofectamine RNAiMAX Transfection Reagent (Thermo Fisher Scientific, Waltham, MA), as we previously described (Zhang et al, 2017).

### Cell viability assays

Cell viability assays were conducted with the CellTiter-Glo Luminescent Assay (Promega, Fitchburg, WI) after 72 h of exposure to serial fold drug dilutions, as described previously (Ferrarotto et al, 2016; Wang et al, 2016). Drug synergy between volasertib and tepotinib was determined by the combination index, which was generated according to the Chou–Talalay median effect method (Chou, 2010) using the CalcuSyn software (Biosoft, Cambridge, UK).

### Colony formation assays

For clonogenic assays, cells were treated for 24 h with dimethyl sulfoxide, volasertib, tepotinib, or the combination of volasertib and tepotinib and then incubated in drug-free medium for 14–21 days. At the end of the assay, cells were washed, fixed, and stained with crystal violet as described previously (Kalu et al, 2018). The total number of colonies per well was estimated using the ImageJ software program (National Institutes of Health, Bethesda, MD). All clonogenic assays were performed in triplicate, and each test was completed twice on different days.

### Reverse phase protein array

Reverse phase protein array was used to compare the expression of 301 proteins (Kalu et al, 2017) using techniques we recently described (Mazumdar et al, 2014), in three parental epithelial NSCLC cell lines, three TGF-β–induced isogenic mesenchymal NSCLC cell lines, and two mesenchymal NSCLC cell lines after incubation with 50 nM volasertib or vehicle for 24 h. The experiments were performed in triplicate on three different days.

### Western blot analysis

Equal amounts of NSCLC cell lysates were separated using 4–20% SDS–PAGE, transferred, and immunoblotted with the indicated primary antibodies, and detected using a horseradish peroxidase–conjugated secondary antibody and an enhanced chemiluminescence reagent, as described previously (Wang et al, 2016). Densitometry quantification of the bands was performed with the ImageJ software program (National Institutes of Health).

### Immunoprecipitation assay

Immunoprecipitation was done with Magnetic Dynabeads (Thermo Fisher Scientific, Waltham, MA) according to the manufacturer's protocol. Briefly, control and treated cell lysates were incubated with the indicated antibody–Dynabeads conjugate followed by purification of antigen–antibody–Dynabeads conjugate and separation on SDS–PAGE for immunoblotting.

### Apoptosis analyses

Apoptosis was measured by TUNEL (terminal deoxynucleotidyl transferase dUTP nick end labeling) staining (Apo-BrdU Kit; BD Biosciences, San Jose, CA) as described previously (Kalu et al, 2017; Wang et al, 2016). Data were acquired using a flow cytometer (Gallios; Beckman Coulter, Brea, CA) and analyzed using Kaluza software (Beckman Coulter, Brea, CA).

### Comet assay

The comet assay was done to detect DNA fragmentation, according to the manufacturer's instructions (Trevigen, Gaithersburg, MD) and as we previously described (Peng et al, 2016; Sen et al, 2012). After fixation, lysis and electrophoresis slides were stained with Vista Green for fluorescence imaging. For evaluation of the comet patterns, about 100 nuclei from each slide were analyzed by Comet Score Pro (TriTek Corp., Sumerduck, VA), and tail moment was calculated by multiplying tail DNA percentage by the length of the tail (Wang et al, 2016).

### Quantitative PCR

Total cellular RNA was isolated using the RNeasy Mini Kit (Qiagen, Hilden, Germany), and complementary DNAs (cDNAs) were synthesized using iScript Reverse Transcription Supermix (Bio-Rad Laboratories, Hercules, CA). Quantitative real-time PCR assays were done with iTaq Universal SYBR Green Supermix (Bio-Rad Laboratories) and the CFX96 Real-time System (Bio-Rad Laboratories) as described previously (Kalu et al, 2017). Primers are listed in Appendix Table S4.

### HGF estimation

Secretory HGF was estimated using the HGF ELISA Kit (Abcam, Cambridge, MA) according to the manufacturer's protocol. Briefly, NSCLC cell lines were treated with 50 nM volasertib in serum-free medium for 24 h, and HGF levels were measured in conditioned medium.

### Subcutaneous xenograft animal models

In conducting research using animals, the investigators adhered to the laws of the United States and regulations of the US Department of Agriculture. All animal experiments were reviewed and approved

**The paper explained**

**Problem**

Most NSCLC remains incurable despite survival benefits from targeted and immunotherapy. Thus, there is still an urgent need for effective systemic therapy. While Plk1 inhibitors are well tolerated by patients and some striking clinical responses were observed, response rates were generally low. Likewise, Plk1 inhibitors lead to diverse biological effects in cancer cells. Understanding the basis for Plk1 inhibitor–induced apoptosis is essential to maximizing their antitumor efficacy. To address this need, our laboratory previously discovered that mesenchymal NSCLC cell lines are more sensitive to Plk1 inhibitors than epithelial cell lines *in vitro* and *in vivo*. However, mechanisms of resistance to Plk1 inhibitors have not been elucidated and this unknown is a major gap in knowledge.

**Results**

We used isogenic pairs of epithelial and mesenchymal NSCLC cell lines to measure changes in the expression of 301 proteins after Plk1 inhibition. We observed differential regulation of the cMet/FAK/Src axis, which is intact in both mesenchymal and epithelial cells. However, Plk1 inhibition inhibits cMet phosphorylation only in mesenchymal NSCLC cells, leading to subsequent inhibition of FAK and Src. Constitutively active cMet abrogates Plk1 inhibitor–induced apoptosis. Likewise, cMet silencing or inhibition enhances Plk1 inhibitor–induced apoptosis. Additionally, cells with acquired resistance to Plk1 inhibitors are more epithelial than their parental cells and maintain cMet activation after Plk1 inhibition. In both patient-derived and cell line xenografts, mesenchymal NSCLC was more sensitive to Plk1 inhibition alone than was epithelial NSCLC. The combination of cMet and Plk1 inhibition led to regression of tumors in three models and marked tumor size reduction in the fourth model. When drug treatment was stopped, tumors treated with the combination did not regrow. Plk1 regulates cMet via the vimentin protein that is only expressed in mesenchymal NSCLC.

**Impact**

NSCLC cell lines have diverse sensitivities to Plk1 inhibition, which is consistent with the results of clinical trials of Plk1 inhibitors in solid tumors. This study reveals a novel mechanism of non-canonical cMet activation in resistant/epithelial NSCLC after Plk1 inhibition. The addition of cMet inhibitors is a promising therapeutic strategy to overcome *de novo* and acquired resistance to Plk1 inhibitors in patients with NSCLC.

by the Institutional Animal Care and Use Committee (IACUC) at The University of Texas MD Anderson Cancer Center and conducted in accordance with MD Anderson's Office of Research Administration and IACUC guidelines. Two million NSCLC cells per mouse were injected subcutaneously, or logarithmically growing PDXs (Hao *et al*, 2015) were implanted subcutaneously into the flanks of 6- to 8-week-old female nude mice (Harlan Laboratories, Indianapolis, IN). Tumors were measured by caliper twice weekly by two, non-blinded, independent investigators. Tumor volumes were calculated as follows: (length × width$^2$)/2. When tumor volumes reached 150 mm$^3$, the mice were randomized and treated with 30 mg/kg volasertib intravenously weekly, 25 mg/kg tepotinib daily (5 days on and 2 days off) by oral gavage, both agents, or vehicle control for 5 weeks. Mice were then euthanized and tumors excised.

A linear model was fit to the data using generalized least squares with two variables (treatment and day) and an autocorrelation structure of order 1 in residuals, given that we had a single grouping variable of mouse identification. The Tukey test was used for pairwise comparisons between treatment groups, and therefore, *P*-values corrected for multiple comparisons were produced, as described previously (Zhang *et al*, 2017). Log-rank tests were conducted to assess the association of the treatment type with survival. Benjamini–Hochberg correction was applied to *P*-values from pairwise comparisons between treatment groups to adjust for multiple comparisons. In addition, Kaplan–Meier curves were generated.

**TUNEL tissue staining**

Harvested tumor tissues were fixed in 10% formalin, embedded in paraffin, and sectioned at 5 μM. Briefly, after deparaffinization and rehydration, tissue sections were subsequently processed for DNA labeling by TdT dNTP followed by Strep-HRP/DAB detection using the TUNEL Apoptosis Detection Kit (Trevigen, Gaithersburg, MD). Images were taken at 40× magnification, and image analysis was performed to calculate the percentage of TUNEL-positive cells using ImageJ software (National Institutes of Health).

**Statistical analysis**

All statistical analyses were performed using R packages (https://www.r-project.org/), a publically available and widely used statistical computing tool. RPPA protein expression and Plk1 inhibitor (BI2536, GSK461364, BRD-K70511574, and GW-843682X) sensitivity data were downloaded from the MD Anderson Cell Line Project (http://tcpaportal.org/mclp/#/) and CTRPv2 (Broad Institute; https://portals.broadinstitute.org/ctrp/) databases and curated for NSCLC cell lines on March 5, 2017. RPPA data for 71 cell lines were available for BI2536, 69 cell lines for GSK461364, 68 cell lines for BRD-K70511574, and 64 cell lines for GW-843682X. To identify correlation between drug sensitivity and protein expression, we applied Spearman's correlation using R. For RPPA proteins to be compared between cell lines after treatment with a Plk1 inhibitor, it is necessary to treat cell line as a random effect because the cell line variation is dominant in principal component analysis. Further contrasts were conducted to compare changes in epithelial cells after treatment with the Plk1 inhibitor, changes in mesenchymal cells after treatment with the Plk1 inhibitor, and changes between the two, which is the interaction term. The analysis was carried out by the limma package in R. To correct for multiple hypothesis testing, we adjusted the resulting *P*-values using the Benjamini–Hochberg method.

There are a variety of group comparisons with respect to different study designs, including two-factor and three-factor factorial designs. All *in vitro* experiments were repeated at least twice (biological replicates) with three or more technical replicates. Analysis of variance was applied, and various contrasts were set up to test the significance of difference between groups of interest and of the interaction of two factors. Bonferroni or Benjamini–Hochberg correction was applied to adjust for multiple testing. For tumor growth curve analysis, linear modeling using generalized least squares with two variables (treatment and day) and an autocorrelation structure of order 1 in residuals were applied. The Tukey test was used for pairwise comparisons among treatment groups, and *P*-values corrected for multiple comparisons were produced. For

survival analysis, log-rank tests were conducted to assess the association of the treatment type with survival. Benjamini–Hochberg correction was applied to *P*-values from pairwise comparisons between treatment groups to adjust for multiple comparisons. In addition, Kaplan–Meier curves were generated. A summary of statistical methods for each specific Figure is provided in Appendix Table S5.

**Expanded View** for this article is available online.

## Acknowledgements

We thank Erica Goodoff of the Department of Scientific Publications at MD Anderson for editing this manuscript. Flow cytometry, bioinformatics, and animal facilities are supported by the National Cancer Institute Cancer Center Support Grant P30CA016672. PDX generation and annotation were supported by philanthropic contributions to The University of Texas MD Anderson Cancer Center Lung Moon Shot Program, Specialized Program of Research Excellence (SPORE) grant CA070907, and University of Texas PDX Development and Trial Center grant U54CA224065. The research was supported by generous donations and by the Office of the Assistant Secretary of Defense for Health Affairs through the Lung Cancer Research Program, under Award No. W81XWH-17-1-0206 (F.M.J.). Opinions, interpretations, conclusions, and recommendations are those of the authors and are not necessarily endorsed by the Department of Defense.

## Author contributions

RS designed and performed most of the experiments and wrote the first draft of the manuscript. SP, PV, and VS assisted with experimental design and conduct of the animal experiments. BF provided and characterized several PDX models. LS, XR, and JW performed the bioinformatics and statistical analyses. FMJ oversaw the study design and wrote the final draft of the paper.

## Conflict of interest

Faye M. Johnson has received research funding from PIQUR Therapeutics and Trovagene. Other authors have no conflicts of interest to declare.

## For more information

https://www.mdanderson.org/research/departments-labs-institutes/labs/faye-johnson-laboratory.html

https://portals.broadinstitute.org/ccle

https://tcpaportal.org/mclp/#/

https://portals.broadinstitute.org/ctrp/

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
