## [Review Process File · EMBO Molecular Medicine]

Noncanonical cMet regulation by vimentin mediates Plk1 inhibitor-induced apoptosis

Ratnakar Singh, Shaohua Peng, Pavitra Viswanath, Vaishnavi Sambandam, Li Shen, Xiayu Rao, Bingliang Fang, Jing Wang, Faye M. Johnson

Review timeline:	Submission date:	19 October 2018
	Editorial Decision:	10 December 2018
	Revision received:	11 February 2019
	Editorial Decision:	25 February 2019
	Revision received:	28 February 2019
	Accepted:	12 March 2019

Editor: Céline Carret

Transaction Report:

1st Editorial Decision

10 December 2018

Thank you for the submission of your manuscript to EMBO Molecular Medicine. We have now heard back from the two referees whom we asked to evaluate your manuscript.

You will see from the set of comments attached that both reports are supportive. Some mechanistic links have to be tightened, and referee 2 recommends amending the discussion. Please also have a careful look at the abstract, in our experience, a clear and concise abstract is much more attractive to the readers.

We would welcome the submission of a revised version within three months for further consideration and would like to encourage you to address all the criticisms raised as suggested to improve conclusiveness and clarity. Please note that EMBO Molecular Medicine strongly supports a single round of revision and that, as acceptance or rejection of the manuscript will depend on another round of review, your responses should be as complete as possible.

I look forward to receiving your revised manuscript.

***** Reviewer's comments *****

Referee #1 (Comments on Novelty/Model System for Author):

See remarks to authors below.

Referee #1 (Remarks for Author):

In the current manuscript, the authors seek to elucidate the mechanism(s) of resistance to PLK inhibitors (PLKi's) in epithelial NSCLC cell lines. In a mostly well conducted and thorough study, they demonstrated that MET was increased in isogenic cell lines after induction of mesenchymal phenotype with TGF beta. As expected these mesenchymal cells lines were more sensitive to PLKi. In addition, PLKi specifically inhibited pMET in the mesenchymal cell lines. Overexpression of MET-TPR induced resistance to PLKi and decreased PLKi induced apoptosis. Interestingly, selection for PLKi resistance cells lead to upregulation of MET as well. Conversely, genetic or pharmacologic inhibition of MET enhanced sensitivity to PLKi in both epithelial and mesenchymal cell lines. The combination of METi and PLKi had significant in vivo activity. Finally, the authors found that PLK1 regulates MET activity through a ligand-independent Beta1-intergrin pathway. This well conducted study could be improved if the following concerns were addressed.

1. The authors claim that PLK1 activates MET through a ligand independent bBeta1-integrin pathway based on the previous study (Jahangiri PNAS 2017) which reported that the MET activation by the Beta1-integrin pathway occurred independent of HGF and their current data demonstrated that this b1-integrin pathway is modulated by PLKi. It would be helpful to support the claim that PLK activates MET primarily through a ligand independent pathway by demonstrated that inhibition of HGF (anti-HGF) does not significantly change pMET in these cells nor further decreases pMET in the presence of PLKi.

2. The authors report that high MET expression was correlated with resistance to 4 PLK1 inhibitors. Did the presence of clinically relevant MET mutations (exon 14 skipping) and/or MET amplification in the cell lines examined correlate with resistance as well? Is MET activity dependent on the bBeta1-integrin pathway in this MET altered cell lines? Can PLK inhibitors decrease pMET in MET altered cell lines?

2. It would be helpful if the author could provide some rationale why two EGFR mutant NSCLC and one DDR2 mutant cell line were selected for the initial experiments in Figure 2.

3. Fig 2A, given the modest change in E-Cadherin expression after TGF-Betab, it would be helpful to see some protein levels of some of their other epithelial and mesenchymal markers in addition to mRNA levels.

4. Fig 2B-C, can the authors explain why there is such a dramatic difference in the relative apoptosis between Volasertib in parental vs. TGF-Betab-treated cells depending on the assay used (cleaved PARP vs. BrdU positive cells) especially for the HCC366 cell line? Since the second assay was a TUNEL based assay could some of the difference simply be explained by increased DNA damage leading to increase labeling of free DNA ends?

4. Figure 3D, lighter exposures of the several of these proteins should be shown including pMET, pFAK and Src as these overexposed blots obscure differences after volasertib.

Minor:

1. Please fix minor typo in abstract line 3 "resistance to in"

Referee #2 (Comments on Novelty/Model System for Author):

This manuscript presents a comprehensive study of the molecular mechanisms driving sensitivity of mesenchymal cells to Plk1 inhibitors, and presents convincing in vivo results of a novel combination therapy to treat mesenchymal and epithelial NSCLC. Generally, the experiments are well designed and executed. However, certain aspects of the analysis of the Met/FAK/Scr cascade need further clarification, and the final conclusion regarding c-Met's role as a driver of resistance would need to be discussed from a different perspective. Combination therapy of Plk1 and c-Met inhibitors is not

novel and it has recently been suggested and demonstrated in vitro (Narvaez et al., 2017). That is why I have graded the novelty as medium. If the authors provide more conclusive results on the above points after major revisions, I would recommend accepting this manuscript for publication, given the potential biological and clinical relevance of this study.

Referee #2 (Remarks for Author):

In this manuscript entitled "Noncanonical cMet activation mediates resistance to Plk1 inhibitor-induced apoptosis" and to be published in EMBO Molecular Medicine, Singh et al. investigate the molecular mechanisms underlying resistance to Plk1 inhibition in non-small cell lung cancer (NSCLC) using both in vitro and in vivo approaches. The authors show that c-Met expression is positively correlated with epithelial-like traits and Plk1-inhibitor resistance, and that its expression is differentially regulated in epithelial and mesenchymal cell lines. Singh and colleagues demonstrate that c-Met activation also decreases Plk1 inhibitor-induced apoptosis. The authors confirmed that mesenchymal NSCLC xenografts responded more effectively to Plk1 inhibition compared to epithelial NSCLC xenografts, and that combination therapy using both Plk1 and c-Met inhibitors resulted in tumour regression. They further uncover the underlying mechanisms linking Plk1 inhibition and c-Met regulation. This manuscript therefore presents a novel mechanism that explains mesenchymal-Plk1 inhibitor sensitivity. Although combination therapy of Plk1 and c-Met inhibitors is not novel and it has recently been suggested and demonstrated in vitro (Narvaez et al, 2017), the authors demonstrate for the first time the molecular mechanisms of sensitivity to a Plk1 inhibitor, presenting in vitro and in vivo evidence of a relevant and interesting finding that can contribute to the formulation of predictive biomarkers for the selection of patients likely to respond to this type of inhibitors in the clinic.

Major concerns:

1. A main concern is the use of volasertib (BI 6727) in this study. This Plk1 inhibitor was not included in the initial protein expression analysis, and it has been reported not to be completely selective and rather act as a non-specific alkylator, also inhibiting Plk2 and Plk3 with similar IC50 values (Gjertsen et al, 2015; Rudolph et al, 2009). Why did the authors decide to use this inhibitor in the study?
2. In my view, the cMet/FAK/Scr cascade analysis in Figure 3D and 3E is weak and, presented on its own, requires additional supporting evidence and validation.
 - 2.1 Given the reported unspecificity of volasertib, it would be crucial to present the results from Plk1 silencing alongside to confirm selectivity of the drug. Si-PLK1 effect on the expression of the cascade factors would therefore be important to support these results.
 - 2.2 According to the authors, Plk1 inhibition inhibits cMet phosphorylation only in mesenchymal cell lines, which results in subsequent inhibition of FAK and Src. However, western blotting in Figure 3E shows that FAK phosphorylation is markedly inhibited in epithelial H1975 cells following Plk1 inhibition, while it is quickly recovered and highly activated in mesenchymal cells, rendering the conclusions about this cascade dubious. Can the authors give an explanation or provide supporting evidence? Perhaps densitometric quantification would be useful as levels of phosphorylation inhibition, especially for FAK and Src, cannot really be compared between epithelial/mesenchymal cells if not normalized to control values.
 - 2.3 The authors show that epithelial cell lines are resistant to Plk1 inhibitors, however in Figure 3E they show that in H1975 epithelial cell line volasertib treatment, phosphorylation of Plk1 substrate NPM1 was inhibited. How do the authors explain this?
 - 2.4 I believe the analysis of the molecular mechanisms underlying Plk1 inhibition would be better presented as a combination/reorganization of the current Figures 3 and 8, which would certainly make the study more convincing.
3. In vivo results are impressive and striking, supporting previous in vitro conclusions. However, it would have been more informative to analyze the effect that the combination therapy (Plk1+c-Met

inhibitors) has in the long term in epithelial tumors rather than in mesenchymal, as these seem to be the most problematic in the clinic. Could the authors provide any data on epithelial tumor growth during recovery phase?

4. Given the final conclusions on the molecular mechanisms underlying sensitivity to Plk1 inhibitor, despite c-Met seems to be more expressed in epithelial tumor cells, it seems delusive to state that c-Met is the responsible for mediating resistance to Plk1-inhibitor-induced apoptosis. Vimentin providing sensitivity to the drug in mesenchymal cells would be a better-founded conclusion.

Minor points:

1. Figure reference to correlation between cMet protein expression and sensitivity to different Plk1 inhibitors in first subsection of Results is incomplete: Fig EV1E.

2. The wording of Figure 1 legend and its respective part in the Results section is confusing: 'Spearman correlations between proteins and sensitivity to Plk1 inhibitors', 'We also found that cMet protein expression correlated with drug sensitivity for all Plk1 inhibitors'. As per the figure content, positive coefficients show a positive association with resistance, not sensitivity, to the inhibitors, as well as cMet protein expression correlates with resistance. Please make these changes to prevent confusions or contradictions.

3. In Figure 3D, + and - symbols are misplaced and not located above their corresponding lane. Please correct this for clearer visualisation of western blot bands.

4. The title of the third subsection of Results states "Activation of cMet, FAK, and Src is differentially regulated in epithelial and mesenchymal NSCLC cell lines following Plk1 inhibition and knockdown", however the knockdown of Plk1 does not seem to be included in this part of the Results, although it would certainly be very informative if included here.

1st Revision - authors' response

11 February 2019

Referee #1 states that "this well conducted study could be improved if the following concerns were addressed:"

1. The authors claim that Plk1 activates MET through a ligand independent β 1-integrin pathway based on the previous study (Jahangiri PNAS 2017) which reported that the MET activation by the β 1-integrin pathway occurred independent of HGF and their current data demonstrated that this β 1-integrin pathway is modulated by PLKi. It would be helpful to support the claim that PLK activates MET primarily through a ligand independent pathway by demonstrated that inhibition of HGF (anti-HGF) does not significantly change pMET in these cells nor further decreases pMET in the presence of PLKi.

We tested the effect of an HGF blocking antibody on cMet activation in two mesenchymal NSCLC cell lines (Calu6 and H1792) and added these data as Figure EV5 B. The HGF neutralizing antibody did not affect basal c-Met activation or Plk1 inhibition-induced cMet inhibition in Calu6. In a second mesenchymal NSCLC cell line (H1792), the HGF neutralizing antibody did reduce basal c-Met activation consistent with cancer cell production of HGF leading to activation c-Met that has previously been well established in some NSCLC tumors (Salgia, 2017). The combination of the HGF neutralizing antibody and Plk1 inhibition had an additive effect on cMet activation (Fig EV5 B).

In the discussion we now note that: "Our data suggest that vimentin-dependent cMet activation is independent of canonical mechanisms of cMet activation because an HGF neutralizing antibody can still affect cMet activation in mesenchymal NSCLC and because robust cMet activation by an activating mutation or amplification was not affected by Plk1 inhibition."

2. The authors report that high MET expression was correlated with resistance to 4 Plk1 inhibitors. Did the presence of clinically relevant MET mutations (exon 14 skipping) and/or MET amplification

in the cell lines examined correlate with resistance as well? Is MET activity dependent on the β 1-integrin pathway in this MET altered cell lines? Can PLK inhibitors decrease pMET in MET altered cell lines?

We previously examined the correlation between mutations of 264 genes that are commonly mutated in cancer, including *MET*, and the drug sensitivity of 63 NSCLC cell lines treated with BI2536 or volasertib or GSK461364 (Ferrarotto et al, 2016). Mutations in *MET* did not correlate with sensitivity to these three Plk1 inhibitors. However, only one NSCLC cell line in the analysis had an activating mutation in exon 14 of *MET* making it impossible to determine if this molecular subgroup was more resistant to Plk1 inhibition although that cell line is resistant to Plk inhibition. We added this information to the introduction.

Plk1 inhibitor sensitivity did not correlate with *MET* amplification in NSCLC cell lines and this new analysis was added on page 5. However, this analysis was limited by the fact that there was drug sensitivity data for only two NSCLC cell lines with *MET* copy number greater than 5.

To answer the question regarding the Plk1- β 1 integrin-cMet pathway in *MET* altered cell lines, we used H596 that has an exon 14 skipping mutation and mesenchymal NSCLC cell lines with varying *MET* copy number. These data were added as Figure 6D.

3. It would be helpful if the author could provide some rationale why two *EGFR* mutant NSCLC and one *DDR2* mutant cell line were selected for the initial experiments in Figure 2.

These cell lines were chosen because they are epithelial NSCLC cell lines that are universally resistant to all Plk1 inhibitors tested. In our prior published research, we determined that none of 264 genes that are commonly mutated in cancer correlated robustly with sensitivity to three different Plk1 inhibitors although *RAS* mutant (*HRAS+KRAS*) NSCLC cell lines were modestly more sensitive (Ferrarotto et al, 2016). Thus, the cell lines in Figure 2 were chosen independently of mutation status. This rationale was added on page 6.

4. Fig 2A, given the modest change in E-Cadherin expression after TGF- β treatment, it would be helpful to see some protein levels of some of their other epithelial and mesenchymal markers in addition to mRNA levels.

As suggested, we added mRNA levels for 6 genes involved in EMT and additional proteins to the Western blot for Figure 2A.

5. Fig 2B-C, can the authors explain why there is such a dramatic difference in the relative apoptosis between Volasertib in parental vs. TGF- β treated cells depending on the assay used (cleaved PARP vs. BrdU positive cells) especially for the HCC366 cell line? Since the second assay was a TUNEL based assay could some of the difference simply be explained by increased DNA damage leading to increased labeling of free DNA ends?

We agree that the difference may be explained by increased DNA damage and timing of the assays (Sundquist et al, 2006). Plk1 inhibition leads to DNA damage (Driscoll et al, 2014; Wang et al, 2018; Yim & Erikson, 2009) in some cancer models, including NSCLC cell lines (Wang et al, 2016) (Fig. 2D). The TUNEL assay is based on free DNA end labeling. In contrast, PARP-1 is specifically proteolyzed by caspase-3 to a 24 kDa DNA-binding domain (DBD) and to a 89 kDa catalytic fragment during the execution of the apoptotic program. The N-terminal p24 fragment is able to bind DNA ends irreversibly and inhibit DNA repair.

We added this information to page 6.

6. Figure 3D, lighter exposures of the several of these proteins should be shown including pMET, pFAK and Src as these overexposed blots obscure differences after volasertib.

Lighter exposures are designated as LE and are shown directly below the higher exposures (HE) for pMet and pFAK. For Src we have replaced the blot with lighter exposure.

Minor: 1. Please fix minor typo in abstract line 3 "resistance to in"

Thank you for pointing out this typo that has been corrected.

Referee #2 (Comments on Novelty/Model System for Author): This manuscript presents a comprehensive study of the molecular mechanisms driving sensitivity of mesenchymal cells to Plk1 inhibitors, and presents convincing *in vivo* results of a novel combination therapy to treat mesenchymal and epithelial NSCLC. Generally, the experiments are well designed and executed. However, certain aspects of the analysis of the Met/FAK/Scr cascade need further clarification, and the final conclusion regarding c-Met's role as a driver of resistance would need to be discussed from a different perspective. Combination therapy of Plk1 and c-Met inhibitors is not novel and it has recently been suggested and demonstrated *in vitro* (Narvaez et al., 2017). That is why I have graded the novelty as medium. If the authors provide more conclusive results on the above points after major revisions, I would recommend accepting this manuscript for publication, given the potential biological and clinical relevance of this study.

We agree that the paper by Narvaez *et. al.* is an excellent paper that supports our findings. We have added these data as a paragraph in the discussion section. As you know, Narvaez *et. al.* developed a drug (Poloppin) that inhibits phosphopeptide binding by Plk1 polo-binding domain (PBD) leading to cell death in *KRAS* mutant cancer cells *in vitro* and decreased tumor size *in vivo* (Narvaez et al, 2017). Because c-Met was upregulated in a colon cancer cell line (SW48) expressing *KRAS*G12D they investigated the combination of *MET* knockdown and c-Met inhibition with Poloppin. cMet inhibition and knockdown did sensitize five *KRAS* mutant cell lines and one pancreatic organoid to Poloppin *in vitro*. No mechanistic data are included in this publication.

We also agree with the reviewer that although Narvaez *et. al.* showed this combination is effective, we “demonstrate for the first time the molecular mechanisms of sensitivity to a Plk1 inhibitor, presenting *in vitro* and *in vivo* evidence of a relevant and interesting finding that can contribute to the formulation of predictive biomarkers for the selection of patients likely to respond to this type of inhibitors in the clinic.” Therefore, we think our research warrants publication with full acknowledgment of the prior study.

Referee #2 (Remarks for Author): 1. A main concern is the use of volasertib (BI 6727) in this study. This Plk1 inhibitor was not included in the initial protein expression analysis, and it has been reported not to be completely selective and rather act as a non-specific alkylator, also inhibiting Plk2 and Plk3 with similar IC50 values (Gjertsen et al, 2015; Rudolph et al, 2009). Why did the authors decide to use this inhibitor in the study?

Volasertib inhibits Plk1, Plk2, and Plk3 at IC50 values of 0.87, 5, and 56 nM, respectively using *in vitro* kinase assays (i.e., cell free biochemical assays) (Rudolph et al, 2009). Although it is not the most specific Plk1 inhibitor available, we chose to use it based on its favorable pharmacokinetics in humans and in mice as well as its clinical relevance at the start of this research. Notably, we used a more specific Plk1 inhibitor (GSK461364) previously and it was also more effective in mesenchymal NSCLC (Ferrarotto et al, 2016).

2. In my view, the cMet/FAK/Scr cascade analysis in Figure 3D and 3E is weak and, presented on its own, requires additional supporting evidence and validation.

2.1 Given the reported unspecificity of volasertib, it would be crucial to present the results from Plk1 silencing alongside to confirm selectivity of the drug. Si-Plk1 effect on the expression of the cascade factors would therefore be important to support these results.

To address issues of specificity, we used low doses of volasertib (< 50nM) and we conducted key experiments with Plk1 knock down. Additional knock down data has been added as Figure 3D.

2.2 According to the authors, Plk1 inhibition inhibits cMet phosphorylation only in mesenchymal cell lines, which results in subsequent inhibition of FAK and Src. However, western blotting in Figure 3E shows that FAK phosphorylation is markedly inhibited in epithelial H1975 cells following Plk1 inhibition, while it is quickly recovered and highly activated in mesenchymal cells, rendering the conclusions about this cascade dubious. Can the authors give an explanation or provide supporting evidence? Perhaps densitometric quantification would be useful as levels of

phosphorylation inhibition, especially for FAK and Src, cannot really be compared between epithelial/mesenchymal cells if not normalized to control values.

In reviewing all the data from the Plk1 inhibition and knock down (Figures 3C-E, EV2A), it is clear that Plk1 inhibition or knock down consistently leads to decreased pMet only in the mesenchymal NSCLC cell lines and not in the epithelial NSCLC cell lines. The reviewer is correct in pointing out that the effects of Plk1 inhibition/knockdown on pSrc and pFAK are more variable in the epithelial cell lines.

We agree that the data do not support the conclusion that “Plk1 inhibition leads to sequential inhibition of cMet and FAK in mesenchymal NSCLC” and we significantly edited this section.

2.3 The authors show that epithelial cell lines are resistant to Plk1 inhibitors, however in Figure 3E they show that in H1975 epithelial cell line volasertib treatment, phosphorylation of Plk1 substrate NPM1 was inhibited. How do the authors explain this?

The Plk1 inhibitors do inhibit Plk1 effectively in all NSCLC cell lines tested independently of drug sensitivity as we previously reported (Ferrarotto et al, 2016). However, our data support a model in which Plk1 inhibition inhibits cMet phosphorylation only in mesenchymal NSCLC lines. Apoptosis in mesenchymal NSCLC is due to both direct effects on Plk1, as well as parallel vimentin and β 1-integrin-mediated, ligand-independent inhibition of cMet. We clarified this information in the introduction, results and discussion.

2.4 I believe the analysis of the molecular mechanisms underlying Plk1 inhibition would be better presented as a combination/reorganization of the current Figures 3 and 8, which would certainly make the study more convincing.

We agree and these changes have been made.

3. In vivo results are impressive and striking, supporting previous in vitro conclusions. However, it would have been more informative to analyze the effect that the combination therapy (Plk1+c-Met inhibitors) has in the long term in epithelial tumors rather than in mesenchymal, as these seem to be the most problematic in the clinic. Could the authors provide any data on epithelial tumor growth during recovery phase?

We do not have any data on the epithelial tumors in the recovery phase. However, mesenchymal NSCLC is more problematic than epithelial NSCLC clinically. EMT is one mechanism leading to loss of oncogene addiction (Thomson et al, 2005). Mesenchymal NSCLC cell lines are significantly more resistant to EGFR and PI3K inhibitors (Byers et al, 2013). Chemotherapy resistance is also associated with EMT, highlighting the importance of targeting cancer cells with a mesenchymal phenotype (Arumugam et al, 2009; Singh & Settleman, 2010). In the discussion, we emphasized the need for combination therapy even in mesenchymal NSCLC by pointing out the reversibility of the EMT and that intratumoral heterogeneity may diminish the efficacy of Plk1 inhibitors alone.

4. Given the final conclusions on the molecular mechanisms underlying sensitivity to Plk1 inhibitor, despite c-Met seems to be more expressed in epithelial tumor cells, it seems delusive to state that c-Met is the responsible for mediating resistance to Plk1-inhibitor-induced apoptosis. Vimentin providing sensitivity to the drug in mesenchymal cells would be a better-founded conclusion.

The reviewer correctly points out that our data support a model in which vimentin and β 1-integrin-mediated inhibition of cMet contributes to Plk1 inhibitor-induced apoptosis in mesenchymal NSCLC. We focused on the c-Met component of this pathway because it is an easily drugable target with clinically relevant agents. We have changed the title of the manuscript and emphasis within the text to indicate that vimentin is providing sensitivity to the drug in mesenchymal cells.

Minor points:

1. Figure reference to correlation between cMet protein expression and sensitivity to different Plk1 inhibitors in first subsection of Results is incomplete: Fig EV1E.

Thank you for pointing out this error that has been corrected.

2. The wording of Figure 1 legend and its respective part in the Results section is confusing: 'Spearman correlations between proteins and sensitivity to Plk1 inhibitors', 'We also found that cMet protein expression correlated with drug sensitivity for all Plk1 inhibitors'. As per the figure content, positive coefficients show a positive association with resistance, not sensitivity, to the inhibitors, as well as cMet protein expression correlates with resistance. Please make these changes to prevent confusions or contradictions.

We corrected this figure legend.

3. In Figure 3D, + and - symbols are misplaced and not located above their corresponding lane. Please correct this for clearer visualisation of western blot bands.

We corrected this figure.

4. The title of the third subsection of Results states "Activation of cMet, FAK, and Src is differentially regulated in epithelial and mesenchymal NSCLC cell lines following Plk1 inhibition and knockdown", however the knockdown of Plk1 does not seem to be included in this part of the Results, although it would certainly be very informative if included here.

These data were added to Figure 3C.

References for this letter:

Arumugam T, Ramachandran V, Fournier KF, Wang H, Marquis L, Abbruzzese JL, Gallick GE, Logsdon CD, McConkey DJ, Choi W (2009) Epithelial to mesenchymal transition contributes to drug resistance in pancreatic cancer. *Cancer Res* 69: 5820-5828

Byers LA, Diao L, Wang J, Saintigny P, Girard L, Peyton M, Shen L, Fan Y, Giri U, Tumula PK et al (2013) An epithelial-mesenchymal transition gene signature predicts resistance to EGFR and PI3K inhibitors and identifies Axl as a therapeutic target for overcoming EGFR inhibitor resistance. *Clin Cancer Res* 19: 279-290

Driscoll DL, Chakravarty A, Bowman D, Shinde V, Lasky K, Shi J, Vos T, Stringer B, Amidon B, D'Amore N et al (2014) Plk1 inhibition causes post-mitotic DNA damage and senescence in a range of human tumor cell lines. *PLoS One* 9: e111060

Ferrarotto R, Goonatilake R, Yoo SY, Tong P, Giri U, Peng S, Minna J, Girard L, Wang Y, Wang L et al (2016) Epithelial-Mesenchymal Transition Predicts Polo-Like Kinase 1 Inhibitor-Mediated Apoptosis in Non-Small Cell Lung Cancer. *Clin Cancer Res* 22: 1674-1686

Narvaez AJ, Ber S, Crooks A, Emery A, Hardwick B, Guarino Almeida E, Huggins DJ, Perera D, Roberts-Thomson M, Azzarelli R et al (2017) Modulating Protein-Protein Interactions of the Mitotic Polo-like Kinases to Target Mutant KRAS. *Cell Chem Biol* 24: 1017-1028 e1017

Rudolph D, Steegmaier M, Hoffmann M, Grauert M, Baum A, Quant J, Haslinger C, Garin-Chesa P, Adolf GR (2009) BI 6727, a Polo-like kinase inhibitor with improved pharmacokinetic profile and broad antitumor activity. *Clin Cancer Res* 15: 3094-3102

Salgia R (2017) MET in Lung Cancer: Biomarker Selection Based on Scientific Rationale. *Mol Cancer Ther* 16: 555-565

Singh A, Settleman J (2010) EMT, cancer stem cells and drug resistance: an emerging axis of evil in the war on cancer. *Oncogene* 29: 4741-4751

Sundquist T, Moravec R, Niles A, O'Brien M, Riss T (2006) Timing Your Apoptosis Assays. <https://www.promega.com/resources/pubhub/cellnotes/timing-your-apoptosis-assays/>

Thomson S, Buck E, Petti F, Griffin G, Brown E, Ramnarine N, Iwata KK, Gibson N, Haley JD (2005) Epithelial to mesenchymal transition is a determinant of sensitivity of non-small-cell lung

carcinoma cell lines and xenografts to epidermal growth factor receptor inhibition. *Cancer Res* 65: 9455-9462

Wang Y, Singh R, Wang L, Nilsson M, Goonatilake R, Tong P, Li L, Giri U, Villalobos P, Mino B et al (2016) Polo-like kinase 1 inhibition diminishes acquired resistance to epidermal growth factor receptor inhibition in non-small cell lung cancer with T790M mutations. *Oncotarget* 7: 47998-48010

Wang Y, Wu L, Yao Y, Lu G, Xu L, Zhou J (2018) Polo-like kinase 1 inhibitor BI 6727 induces DNA damage and exerts strong antitumor activity in small cell lung cancer. *Cancer Lett* 436: 1-9
Yim H, Erikson RL (2009) Polo-like kinase 1 depletion induces DNA damage in early S prior to caspase activation. *Mol Cell Biol* 29: 2609-2621

2nd Editorial Decision

25 February 2019

Thank you for the submission of your revised manuscript to EMBO Molecular Medicine. We have now received the enclosed reports from the referees that were asked to re-assess it. As you will see the reviewers are now globally supportive and I am pleased to inform you that we will be able to accept your manuscript pending the following final amendments:

1) Please address the minor text changes commented by referee 2

We are looking forward to receiving your revised article.

***** Reviewer's comments *****

Referee #1 (Comments on Novelty/Model System for Author):

See remarks to authors below.

Referee #1 (Remarks for Author):

The authors have thoroughly addressed my comments. Nice Work!

Referee #2 (Comments on Novelty/Model System for Author):

This paper is very well conducted as per the experimental designs and at the technical level. I have scored the novelty as "Medium" because of the previous findings published in the Narvaez et al (2017) manuscript, although I agree with the authors that no relevant mechanistic data were included in this publication, and the authors now demonstrate for the first time the molecular mechanisms of sensitivity to a Plk1 inhibitor, providing solid in vitro and in vivo data. The potential medical impact is high, due to the poor survival rates of NSCLC patients and current treatment resistance. The models in this study are very convenient, including epithelial and mesenchymal PDXs and NSCLC cell line in vivo models.

Referee #2 (Remarks for Author):

In this manuscript now entitled "Noncanonical Met regulation by vimentin mediates Plk1 inhibitor-induced apoptosis" the authors present a comprehensive and well-conducted study on the molecular mechanisms underpinning sensitivity to Plk1 inhibition in mesenchymal and epithelial non-small cell lung cancer tumours. The study is well designed and includes mechanistic data supporting sensitivity to this type of inhibitors, as well as it addresses its translation into different in vitro and in vivo models, including patient-derived xenografts. After major revisions, the authors have carefully addressed all points previously raised. The additional data included in the manuscript, in particular in Figures 2, 3 and 6, as well as the modifications in the text and plausible explanations added have significantly improved the manuscript, dismissing previous ambiguities or unclear/unsupported conclusions.

The current title now properly reflects the main finding of the paper, and despite combination therapy of Plk1 and c-Met inhibitors has previously been suggested (Narvaez et al, 2017), the findings of this exhaustive study provide full validation of a novel pathway of important biological relevance in NSCLC that can potentially be translated into the clinic. I congratulate the authors for this comprehensive study.

I just have minor suggestions:

1. After the first sentence in the abstract it may not be clear enough for the reader what is specifically found in this study. Whether differential regulation of cMet is part of the previous or the current study seems not obvious as it is stated. I would suggest introducing an additional wording such as "Here, we show..." in order to separate previous findings from novel findings.
2. The following wording when describing Fig 1B in the text can be confusing: "We also found that cMet protein expression correlated with drug sensitivity for all Plk1 inhibitors". Could the authors please make changes to clarify the effect, e.g.: "cMet protein expression correlated with drug resistance" or "cMet protein expression negatively correlated with drug sensitivity".
3. Figure 3D legend. Full p-value reference is missing: " $*p < 0.0$."
4. Figure 6 legend. Could the authors please add p-value for two-asterisk (**) references?

2nd Revision - authors' response

28 February 2019

1) Please address the minor text changes commented by referee 2.

1. After the first sentence in the abstract it may not be clear enough for the reader what is specifically found in this study. Whether differential regulation of cMet is part of the previous or the current study seems not obvious as it is stated. I would suggest introducing an additional wording such as "Here, we show..." in order to separate previous findings from novel findings.

The text was added as suggested.

2. The following wording when describing Fig 1B in the text can be confusing: "We also found that cMet protein expression correlated with drug sensitivity for all Plk1 inhibitors". Could the authors please make changes to clarify the effect, e.g.: "cMet protein expression correlated with drug resistance" or "cMet protein expression negatively correlated with drug sensitivity".

The text was added as suggested.

3. Figure 3D legend. Full p-value reference is missing: " $*p < 0.0$."

This was corrected.

4. Figure 6 legend. Could the authors please add p-value for two-asterisk (**) references?

This was corrected.

Corresponding Author Name: Faye Johnson
Journal Submitted to: EMBO Molecular Medicine
Manuscript Number: EMM-2018-09960